# Hypervolume Maximization: A Geometric View of Pareto Set Learning

**Xiaoyuan Zhang[a], Xi Lin[a], Bo Xue[a], Yifan Chen[b], Qingfu Zhang[a]**
[a] Department of Computer Science, City University of Hong Kong;
City University of Hong Kong Shenzhen Research Institute.
[b] Departments of Mathematics and Computer Science, Hong Kong Baptist University.

## Abstract

This paper presents a novel approach to multiobjective algorithms aimed at modeling the Pareto set using neural networks. Whereas previous methods mainly focused on identifying a finite number of solutions, our approach allows for the direct modeling of the entire Pareto set. Furthermore, we establish an equivalence between learning the complete Pareto set and maximizing the associated hypervolume, which enables the convergence analysis of hypervolume (as a new metric) for Pareto set learning. Specifically, our new analysis framework reveals the connection between the learned Pareto solution and its representation in a polar coordinate system. We evaluate our proposed approach on various benchmark problems and real-world problems, and the encouraging results make it a potentially viable alternative to existing multiobjective algorithms. Code is available at https://github.com/xzhang2523/hvpsl/tree/master.

## 1 Introduction

Pareto solutions [1] effectively balance multiple objectives, making optimal tradeoffs among all objectives. For a Pareto solution, improving one objective without adversely affecting others is not possible. The Pareto set (PS) refers to the set of all Pareto solutions, while the Pareto front (PF) denotes the image of the PS in the objective space. Over the years, numerous methods have been developed to generate a single Pareto optimal solution [2, 3, 4], or a finite set of Pareto solutions [5, 6, 7, 8, 9].

Under mild conditions on continuous optimization problems with $m$ objectives, the Pareto set becomes a $(m\text{-}1)$-D continuous manifold [10, 11]; given that, the finite set learned from classical methods might not accurately approximate the continuous manifold. To overcome this issue, Pareto set learning (PSL) has been recently proposed in [12, 13], with the hope of training a neural network model capable of approximating the *entire* Pareto set. The significant advantage of learning the complete Pareto set is that a trained model can generate the optimal solution for any preference (conceptually the weight of different objectives, see the definition in Lemma 1) in real time, allowing for comprehensive decisions.

The training approach utilized in prior PSL techniques can be viewed as an extension of decomposition-based multiobjective optimization strategies [5] to an infinite preference scenario. However, several unresolved issues existed in prior PSL methods. Firstly, the interpretation of the sum of aggregation functions under different preferences lacks clarity, simply taking all preferences as equally important and thus leading to a partial Pareto front. Secondly, aside from the EPO-based

---

Correspondence to: Yifan Chen <yifanc@hkbu.edu.hk>, Qingfu Zhang <qingfu.zhang@cityu.edu.hk>.

37th Conference on Neural Information Processing Systems (NeurIPS 2023).

PSL proposed in [3], there has been limited investigation into the connection between a preference and its corresponding Pareto solution. Despite the effectiveness of the EPO-based PSL approach, its computational latency has been identified as a significant obstacle in its ability to handle large-scale problems. Thirdly, theoretical guarantees of PSL are lacking, and therefore the underlying mechanism of PSL remains unknown.

After identifying the challenges regarding existing PSL techniques for solving multiobjective optimization problems, we propose a new approach that addresses all these issues. Our method utilizes a geometric perspective on Pareto set learning, which takes the form of a hypervolume maximization problem. This approach brings several benefits. Firstly, it provides a clear interpretation of Pareto set learning as a hypervolume maximization problem, allowing us to build a theoretical bound between the PSL results and the hypervolume, one of the most important indicators in multiobjective optimization. Secondly, our approach establishes a clear correspondence between a specific preference and the resulting Pareto solution in a polar coordinate system. Lastly, the new perspective implies several important techniques in hypervolume-based PSL to learn an entire Pareto set.

The contributions of this work can be summarized as follows:

1. We provide a novel geometric perspective for Pareto set learning, which recognizes its equivalence to hypervolume maximization. As a result, the Pareto solutions derived from this method are aligned precisely with the polar angle under mild conditions, which is the input of the neural network in use.
2. We provide the first theoretical analysis for Pareto set learning, specifically examining the gap between the estimated hypervolume and the true hypervolume of the Pareto set. Our analysis enhances a better understanding of Pareto set learning.
3. We have developed techniques to learn a more complete Pareto front, surpassing previous methods that obtain a partial solution set. Our approach achieves promising results on various benchmark problems and real-world problems.

## 2  Related works

**Multiobjective Optimization via Hypervolume Maximization.** The hypervolume (hv) indicator measures the quality of multiobjective optimization solution sets and is consistent with Pareto dominance. Maximizing hypervolume is a basic principle in multiobjective optimization algorithm design, and several popular algorithms like SMS-EMOA [14], SIBEA [15, 16], and MO-CMA-ES [17] use it to generate a finite number of optimal solutions. A new method named *multiobjective learning using hv maximization* was proposed in [18], which leverages hypervolume maximization while training neural networks to obtain a *finite* set of Pareto solutions. This method differs from our approach as it does not rely on a model to approximate the Pareto set. Instead, it adopts the gradient search method from [19] for hypervolume maximization. A major drawback of this approach is its high computational complexity in obtaining the hypervolume gradient.

**Pareto Set Learning and Conditional Models.** Pareto set learning involves using a neural network conditioned on user preference to learn the entire Pareto set. This concept was first introduced in [12, 13] and has gained popularity in various areas like drug design [20], multitask image classification [21, 22], and multiobjective neural combinatorial optimization [23]. In addition, some studies have used a similar conditional model to generate a set of diverse solutions, including the works addressing multiobjective reinforcement learning [24, 25, 26], and the works generating policies under different conditional levels [27, 28].

## 3  Background

A multiobjective optimization (MOO) problem with $m$ objectives can be stated as follows:

$$\min_{x \in \mathcal{X} \subset \mathbb{R}^n} f(x) = (f_1(x), f_2(x), ..., f_m(x)), \tag{1}$$

where $x$ is the decision variable, $\mathcal{X}$ is a compact decision space, and $f(x) : \mathcal{X} \to \mathbb{R}^m$ is the objective function. The objective domain $\mathcal{Y} = f \circ \mathcal{X}$ [1]. Given two candidate solutions $x^{(a)}, x^{(b)} \in \mathcal{X}$, we

---

[1]The function $f$ applied to set $A$ produces a new set $B$, denoted as $B = \{y | f(x), x \in A\}$.

say $x^{(a)}$ **dominates** $x^{(b)}$ if and only if $\forall\, i \in [m]$ [2], $f_i(x^{(a)}) \leq f_i(x^{(b)})$, and $\exists j \in [m] \setminus i$ such that $f_j(x^{(a)}) < f_j(x^{(b)})$. $x^*$ is a **Pareto solution** if no other solution $x \in \mathcal{X}$ dominates $x^*$; its image $f(x^*)$ is called a **Pareto objective**. The set of all Pareto solutions is called the **Pareto set** (PS). And the image of PS is called the **Pareto front** $\mathcal{T} := f \circ \text{PS}$. In a separate note, a solution $x'$ is called weakly Pareto optimal if there exists no solution $x \in \mathcal{X}$ such that $f_i(x) \prec f_i(x')$ [3]. A Pareto solution is a weakly Pareto solution, but a weakly Pareto solution is not necessarily a Pareto solution.

The nadir/ideal point of a multiobjective problem is constructed by the worst/best objective values of the Pareto set. $y_i^{\text{nadir}} = \sup_{y \in \mathcal{T}}\{y_i\}$, and similarly, $y_i^{\text{ideal}} = \inf_{y \in \mathcal{T}}\{y_i\}$, $\forall i \in [m]$. The hypervolume indicator is defined as a metric of the optimality of a set of objective vectors $A$, detailed as follows.

**Definition 1** (The hypervolume (HV) indicator [29]). *The hypervolume indicator of a set $A$ is defined as:*

$$\mathcal{H}_r(A) := \Lambda(\{q \mid \exists p \in A : p \preceq q \text{ and } q \preceq r\}), \quad (2)$$

*where $\Lambda(\cdot)$ denotes the Lebesgue measure, and $r$ is a reference vector. We require that $r \succeq y^{nadir}$.*

Figure 1 depicts the hypervolume indicator of the Pareto front, represented by the yellow region.

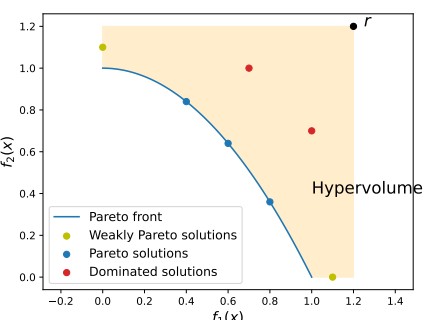

Figure 1: Pareto solutions, weakly Pareto solutions, dominated solutions, and the HV.

# 4 Pareto Set Learning via Hypervolume Maximization

As shown in Figure 2, Pareto set learning (PSL) aims to learn the whole Pareto set by a model, which differs from population-based MOEAs [5, 6]. One straightforward way is to learn a *Pareto neural model* $x_\beta(\cdot) : \Theta \mapsto \mathbb{R}^n$ that translates a polar coordinate $\theta \in \Theta$ into a Pareto solution $x \in \mathbb{R}^n$, where $\Theta = [0, \frac{\pi}{2}]^{m-1}$. After the training, for any $\theta \in \Theta$, the Pareto neural model $x_\beta(\cdot)$ can directly generate an approximate Pareto solution in real-time ($<$1s).

In this paper, we present a novel geometry view to formulate PSL based on hypervolume maximization. Firstly, we notice the following equivalence holds:

**Proposition 1** (Equivalence between PSL and HV maximization). *When $r$ is a reference point dominating the whole Pareto front $\mathcal{T}$, and $A$ is a non-dominated solution set, we have $\mathcal{T} = \arg\max_A \mathcal{H}_r(A)$.*

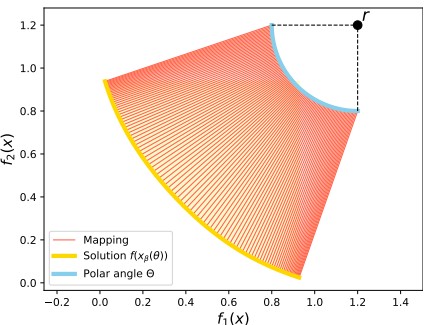

Figure 2: PSL learns the whole Pareto set. For a coordinate $\theta$, model $x_\beta(\cdot)$ learns a corresponding Pareto solution $x_\beta(\theta)$.

Proposition 1 shows that $\mathcal{H}_r(A)$ achieves the maximal value if the model $x_\beta(\cdot)$ learns all Pareto solutions. So, it is natural to convert the Pareto set learning problem with a parameter $\beta$ as a hypervolume maximization problem:

$$\max_\beta \mathcal{H}_r(f \circ x_\beta(\Theta)). \quad (3)$$

In Equation (3), the subscript $r$ refers to a fixed reference point satisfying $y \preceq r, \forall y \in \mathcal{T}$. The objective function measures the quality of the Pareto neural model $x_\beta(\cdot)$ and this objective achieves optimality only when the model learns all Pareto solutions.

## 4.1 Pareto Front Hypervolume as an Expectation

---

[2]$[m]$ denotes the index set $\{1, \ldots, m\}$.

[3]The symbol "$\succeq / \succ$" is used to compare two $m$-D vectors $a$ and $b$. $a \succeq b$ when $a_i \geq b_i, \forall i \in [m]$, and similarly, $a \succ b$ when $a_i > b_i, \forall i \in [m]$.

In this section, we are devoted to casting the optimal solution of Equation (3) (i.e., the hypervolume of the Pareto front) into a polar coordinate form. To accomplish this, we leverage an existing technique to transform the hypervolume calculation of a finite set of solutions into an expectation problem, which is formulated as the following lemma,

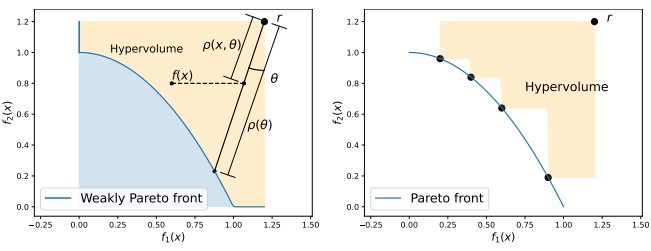

(a) HV of (weakly) Pareto front.    (b) HV of finite points.

Figure 3: Pareto front hypervolume calculation in the polar coordinate. $\rho_{\mathcal{X}}(\theta)$ is the distance from the reference point to the Pareto front along angle $\theta$. $\rho(x, \theta)$ (used in Equation (6)) is the projected distance at angle $\theta$.

**Lemma 1** (Hypervolume scalarization of a finite set [30, 31, 32]). *Let $A = \{y^{(1)}, y^{(2)}, \ldots, y^{(N)}\}$ be set of finite objective vectors and $r$ is a reference point, $r \succeq y^{(i)}$. The hypervolume of set $A$ with $r$ can be expressed in terms of an expectation,*

$$\mathcal{H}_r(A) = c_m \mathbb{E}_\theta \left[ \left( \max_{y \in A} \min_{i \in [m]} \left\{ \frac{r_i - y_i}{\lambda_i(\theta)} \right\} \right)^m \right], \tag{4}$$

*where we specify $\theta$ follows $\mathrm{Unif}(\Theta)$, the uniform distribution on $\Theta$. $\lambda(\theta) \in \mathbb{S}_+^{m-1}$ is the preference vector: it is the Cartesian coordinate of $\theta$ on the positive unit sphere $\mathbb{S}_+^{m-1}$, with $\lambda_1(\theta) = \sin\theta_1 \sin\theta_2 ... \sin\theta_{m-1}$, $\lambda_2(\theta) = \sin\theta_1 \sin\theta_2 ... \cos\theta_{m-1}$, ..., $\lambda_m(\theta) = \cos\theta_1$. $c_m = \frac{\pi^{m/2}}{2^m \Gamma(m/2+1)}$ is a constant that is only dependent on $m$ and $\Gamma(\cdot)$ is the Gamma function.*

We notice that, it is not "economic" to estimate the hypervolume $\mathcal{H}_r(A)$ in Lemma 1: numerous samples from $\mathrm{Unif}(\Theta)$ are needed, while the expensive Monte Carlo estimation only provides the hypervolume for a finite set $A$. Even when $A$ consists of only one point, the calculation of $\mathcal{H}_r(A)$ still relies on the expectation over $\mathrm{Unif}(\Theta)$. However, extending the scope of the input set $A$ in Lemma 1, we discover that the hypervolume of a (weakly) Pareto front containing an infinite number of objective vectors can as well be expressed as an expectation,

$$\mathcal{H}_r(\mathcal{T}) = c_m \mathbb{E}_\theta \left[ \rho_{\mathcal{X}}(\theta)^m \right], \tag{5}$$

with proof provided in Appendix B.2. The distance function denoted as $\rho_{\mathcal{X}}(\theta)$, as illustrated in Figure 3(a), represents the Euclidean distance between a reference point denoted as $r$ and the Pareto front at the coordinate of $\theta$. The distance function $\rho_{\mathcal{X}}(\theta)$ can be precisely evaluated using the equation provided, with the proof available in Appendix B.4.

$$\rho_{\mathcal{X}}(\theta) = \max_{x \in \mathcal{X}} \rho(x, \theta) = \max_{x \in \mathcal{X}} \min_{i \in [m]} \left\{ \frac{r_i - f_i(x)}{\lambda_i(\theta)} \right\}. \tag{6}$$

where $\rho(x, \theta) = \min_{i \in [m]} \left\{ \frac{r_i - f_i(x)}{\lambda_i(\theta)} \right\}$ represents the projected distance of an objective vector $f(x)$ at coordinate $\theta$, as illustrated in Figure 3(a). Let $\lambda$ be the shorthand of $\lambda(\theta)$, we have the following proposition of the optimal solution of Equation (6).

**Proposition 2.** *Given an objective vector $y$ that satisfies $\frac{r_1 - y_1}{\lambda_1} = \ldots = \frac{r_m - y_m}{\lambda_m}$. ①: If $y$ is weakly Pareto optimal, then $y$ is one of the optimal solutions of $\tilde{g}(y, \lambda) = \max_{y \in \mathcal{Y}} \min_{i \in [m]} \left\{ \frac{r_i - y_i}{\lambda_i} \right\}$. ②: If $y$ is Pareto optimal, then $y$ is the only optimal solution of $\tilde{g}(y, \lambda)$.*

*Proof.* ①: If there exists a $y'$ such that $\tilde{g}(y', \lambda) < \tilde{g}(y, \lambda)$, then $y' \prec y$, contradicting the weakly Pareto optimality of $y$. ②: If there exists a $y' \neq y$ such that $\tilde{g}(y', \lambda) = \tilde{g}(y, \lambda)$, then $y' \preceq y$, and there exists at least one index $j$ where $y'_j < y_j$. This contradicts the Pareto optimality of $y$. □

**Remark 1** (On the "exact" Pareto solution). *Proposition 2 establishes a connection between the polar coordinate $\theta$ and the resulting Pareto solution. Once the Pareto neural model is well-trained, and assuming the absence of weakly Pareto solutions and the existence of an exact solution [3], the*

*Pareto neural model can predict the "exact" Pareto solution aligned with the polar angle $\theta$. Since we only utilize an efficient aggregation function (Equation (6)), this property reveals that there is no need to solve the optimization problem (Eq. (24) in [3]) in order to achieve the "exact" Pareto solution.*

We close the subsection with a note on the case of disjointed Pareto fronts. With a disjointed Pareto front $\mathcal{T}$, the function $\rho_{\mathcal{X}}(\theta)$ can serve to measure the distance between the reference point $r$ and the *attainment surface* [33]. In this disjointed scenario, the expectation form of the Pareto front hypervolume by $\rho_{\mathcal{X}}(\theta)$, as specified by Equation (5), still holds (c.f. Appendix B.4).

### 4.2 Alternative Forms of Hypervolume

In this section, we train the Pareto neural model $x_{\beta}(\cdot)$ by maximizing the alternative forms of hypervolumes. The optimization of the value of $\mathcal{H}_r(f \circ x_{\beta}(\Theta))$ as defined in Equation (1) is challenging. However, by utilizing Equation (6), we can choose to instead optimize an easy-to-compute form of hypervolume, denoted by $\overline{\mathcal{H}}_r(\beta)$. This surrogate hypervolume function $\overline{\mathcal{H}}_r(\beta)$, called PSL-HV1, is defined as follows:

$$\overline{\mathcal{H}}_r(\beta) = c_m \mathbb{E}_\theta[\rho_\beta(\theta)], \quad \rho_\beta(\theta) = \begin{cases} \rho(x_\beta(\theta), \theta)^m & \text{if } \rho(x_\beta(\theta), \theta) \geq 0 \\ \rho(x_\beta(\theta), \theta) & \text{otherwise} \end{cases}. \tag{7}$$

Here, $\rho(x_\beta(\theta), \theta) \leq \rho_{\mathcal{X}}(\theta)$ is the projected distance at angle $\theta$ as defined by Equation (6). When the Pareto neural model is supposed to learn the whole Pareto set, $\rho(x_\beta(\theta), \theta) \rightarrow \rho_{\mathcal{X}}(\theta), \forall \theta \in \Theta$ and thus $\overline{\mathcal{H}}_r(\beta) \rightarrow \mathcal{H}_r(\mathcal{T})$.

We also provide a second surrogate hypervolume function, denoted as PSL-HV2 (derivation in Appendix B.5). The Pareto hypervolume can be estimated as the volume difference between the regions dominating $r$ and those that dominate the Pareto front. PSL-HV2 maximizes the following objective,

$$\overline{\mathcal{H}}_r(\beta) = \prod_{i=1}^m (r_i - y_i^{\text{ideal}}) - c_m \mathbb{E}_\theta[\overline{\rho}_\beta(\theta)]. \tag{8}$$

where $\overline{\rho}_\beta(\theta) = \overline{\rho}_{\mathcal{X}}(x_\beta(\theta), \theta)^m$ and $\overline{\rho}_{\mathcal{X}}(\theta)$ is an alternative projected distance function $\overline{\rho}_{\mathcal{X}}(x_\beta(\theta), \theta) = \max_{i \in [m]} \frac{f_i(x_\beta(\theta)) - y_i^{\text{ideal}}}{\lambda_i(\theta)}$. Although the Pareto neural model $x_\beta(\cdot)$ theoretically has the ability to represent the complete Pareto set [34] with both surrogate hypervolume functions, our empirical results show that the quality of the learned solutions is sensitive to the specific choice: Equation (7) and Equation (8) give different performances on various tasks.

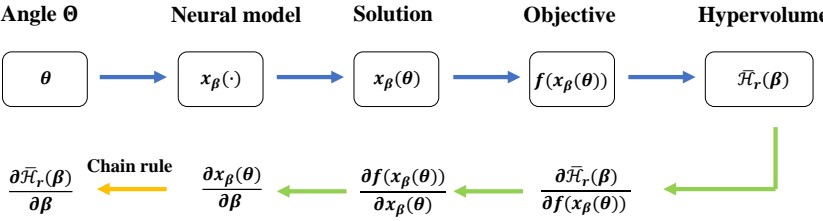

Figure 4: The gradient graph above is used for computing the estimated gradient of the hypervolume.

**Algorithm.** The goal of HV-based PSL is to maximize the objective function, $\max_\beta \overline{\mathcal{H}}_r(\beta)$. Both Equation (7) and Equation (8) can be effectively optimized using gradient descent approaches. The empirical gradient $\nabla_\beta \hat{\overline{\mathcal{H}}}_r(\beta)$ can be obtained efficiently through backpropagation, as illustrated in Figure 4. Their analytical forms are provided in Appendix B.5. For practical algorithms, the *empirical* gradient is estimated from a batch of $N$ angles $\theta^{(1)}, \ldots, \theta^{(N)}$ through sampling from Unif($\Theta$). Then the neural network is updated by $\beta \leftarrow \beta + \eta \nabla_\beta \hat{\overline{\mathcal{H}}}_r(\beta)$, where $\eta$ is a learning rate.

### 4.3 Discussion on Different PSL Approaches

PSL-HV1 and PSL-HV2 have different designs for their projected distance function $\rho(x, \theta)$, with PSL-HV2 exhibiting significant growth as $\lambda$ approaches boundary preferences such as [0,1] or [1,0] (taking the $m = 2$ case as an example). This property brings both pros and cons compared to PSL-HV1. The steep gradient may present numerical difficulties, but it also permits PSL to adopt larger weights and learn boundary solutions more effectively. We have observed that this feature of PSL-HV2 is particularly useful for learning the complete Pareto set, as opposed to previous PSL methods that only learned a partial Pareto front.

To avoid only learning a partial Pareto set, we propose a simple strategy for PSL-HV1 that involves using a larger reference point than the nadir point. This approach, which is common in hypervolume-based optimization methods [35, 36], gives larger weights to boundary preferences and allows for a more complete Pareto set.

Before our study, two primary approaches for PSL optimization were followed [12, 37]: optimizing the expectation using EPO/LS and optimizing using Tchebycheff functions. However, EPO-based PSL is slow for complex multiobjective problems and solely learns a partial Pareto set, and LS-based PSL only finds the convex portion of a Pareto front. Our work extends the approach in [37] by introducing a hypervolume interpretation for PSL. We establish a clear relationship between preferences and solutions and develop a theoretical analysis framework for PSL. Additionally, we propose methods to recover a more complete Pareto set, thus improving upon the existing approaches.

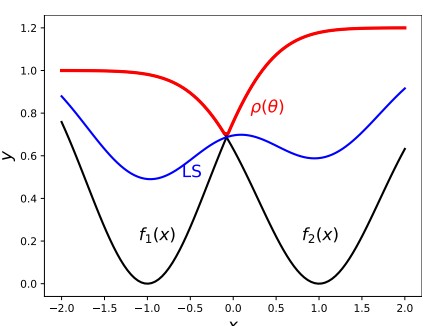

Figure 5: The point-wise maximum of quasi-convex functions preserves quasi-convexity. A positive-weighted linear combination of quasi-convex functions can result in a non-quasi-convex function.

## 5 Statistical Guarantees on Hypervolume Convergence

This section first establishes the bounds for the generalization error of the Pareto front when using the exact distance function $\rho_{\mathcal{X}}(\theta)$ in Section 5.1. Subsequently, we discuss the generalization bound of proposed HV-based Pareto set learning in Section 5.2.

### 5.1 Convergence of the Pareto Front Hypervolume

**Proposition 3.** ①: $\rho(x, \theta)$ is concave when $f_i$'s are convex. ②: $\rho(x, \theta)$ is quasi-concave when $f_i$'s are quasi-convex. ③: $\rho_\beta(\theta)$ defined in Equation (7) is quasi-concave w.r.t $x_\beta(\theta)$ when the objective function $f(x)$ is quasi-convex.

*Proof.* The statements ① and ② are consequences of the preservation of convexity/quasi-convexity by taking the point-wise maximum of convex/quasi-convex functions. The statement ③ is justified by the fact that a non-decreasing composition of a quasi-concave function preserves quasi-concavity. Additionally, it can be observed that the function $f(x) = x^m$ for $x \geq 0$ (and $f(x) = x$ otherwise) is a non-decreasing function. $\square$

**Remark 2.** *To better understand Proposition 3, we analyze a modified bi-objective VLMOP2 problem from [38]. In this example, let $f_1(x) = 1.2(1 - \exp(-(x + 1)^2))$ and $f_2(x) = 1 - \exp(-(x - 1)^2)$. Figure 5 illustrates the limitation of the linear scalarization (LS) approach [39]. It shows that the LS-based PSL may yield multiple local optimas when using a non-negative linear combination of quasi-convex functions $f_1$ and $f_2$, resulting local optimal Pareto models. However, it is worth noting that the function $-\rho(x, \theta)$ considered remains quasi-convex [40].*

Assuming that $b \leq r_i - y_i \leq B, \forall i \in [m], y \in \mathcal{T}$. Let $\mathcal{Z}(\theta) = c_m \rho_{\mathcal{X}}(\theta)^m$, then $\sup_\theta \mathcal{Z}(\theta) \leq c_m B^m m^{m/2}$. Let $\widehat{\mathcal{H}}_r(\mathcal{T}) := \frac{1}{N} \sum_{i=1}^N \mathcal{Z}(\theta^{(i)})$ denote the empirical estimation of $\mathcal{H}_r(\mathcal{T})$ with $N$

samples. Via Hoeffding inequality, similar to [32], we have the following inequality,

$$\Pr(|\widehat{\mathcal{H}}_r(\mathcal{T}) - \mathcal{H}_r(\mathcal{T})| \geq \epsilon) \leq 2\exp\left(\frac{-2N\epsilon^2}{c_m^2 B^{2m} m^m}\right).$$

## 5.2 PSL Generalization Bound

In this section we show that the proposed hypervolume metric can fit into the regular neural network studies and similarly enjoy the statistical guarantees on error $\left|\overline{\mathcal{H}}_r(\beta) - \mathcal{H}_r(\mathcal{T})\right|$ for PSL-HV1. Specifically, the term can be interrupted as the difference between the metric of $x_{\hat{\beta}}$ computed and the metric of ground truth $x^*$, indicating the quality of the MLP map $x_{\hat{\beta}}$. We will shortly show the error above will converge with regular SGD optimizers and large enough sample size $N$, as expected.

We can first decompose the aforementioned error as follows,

$$\left|\overline{\mathcal{H}}_r(\hat{\beta}) - \mathcal{H}_r(\mathcal{T})\right| \leq \underbrace{\left|\overline{\mathcal{H}}_r(\hat{\beta}) - \widehat{\overline{\mathcal{H}}}_r(\hat{\beta})\right|}_{\varepsilon_1} + \underbrace{\left|\widehat{\overline{\mathcal{H}}}_r(\hat{\beta}) - \widehat{\overline{\mathcal{H}}}_r(\beta^*)\right|}_{\varepsilon_2} + $$
$$\underbrace{\left|\widehat{\overline{\mathcal{H}}}_r(\beta^*) - \overline{\mathcal{H}}_r(\beta^*)\right|}_{\varepsilon_3} + \underbrace{\left|\overline{\mathcal{H}}_r(\beta^*) - \mathcal{H}_r(\mathcal{T})\right|}_{\varepsilon_4}, \tag{9}$$

in which we define $\widehat{\overline{\mathcal{H}}}_r(\beta) = \frac{c_m}{N}\sum_{i=1}^{N}[\rho(x_\beta(\theta^{(i)}), \theta^{(i)})^m]$ as the empirical hypervolume estimation associated with parameter $\beta$. $\beta^*$ is a set of reference parameters for the MLP map $x_\beta$. Those errors respectively correspond to generalization error ($\varepsilon_1$), optimization error ($\varepsilon_2$), generalization error ($\varepsilon_3$), and approximation error ($\varepsilon_4$). We can conceptually set the reference parameter $\beta^*$ as a good local maxima (in the local region of $\hat{\beta}$) for hypervolume maximization so that the approximation error $\varepsilon_4$ is small due to the universal approximation character of the MLP model family [41]. In our analysis, we assume that regular SGD technique brings $\hat{\beta}$ close enough to the local maxima $\beta^*$, resulting in a small optimization error $\varepsilon_2$.

The rest steps to show convergence is to control the generalization error $\varepsilon_1$ and $\varepsilon_3$. We present the following theorem to give the uniform convergence rate of the generalization error, which applies to both $\varepsilon_1$ and $\varepsilon_3$.

**Theorem 1** (Generalization error). *Let the object functions $f_i$'s all be $L_f$-Lipschitz, and let $x_\beta$ be an $L$-layer MLP $\sigma_L(W_L\sigma_{L-1}(\cdots\sigma_1(W_1\theta)\cdots))$, with 1-Lipschitz positive homogeneous activation $\sigma_i$'s and $\|W_i\|_F \leq B_w, \forall i \in [L]$ (under this setting $\beta$ is the set of $W_i$'s). The design matrix $X_\theta$ denotes the $N$ collected samples $\theta_i$'s. We further set the reference point $r$ so that $r_i - f_i(x) \in [b, B], \forall i$, for $x \in \mathcal{T} \cup f \circ x_{\beta^*}(\Theta) \cup f \circ x_{\hat{\beta}}(\Theta) \subset \mathbb{R}^n$. With probability at least $1 - \delta$ ($\delta < \frac{1}{2}$), for $\beta = \beta^*$ or $\hat{\beta}$, we have*

$$\left|\overline{\mathcal{H}}_r(\beta) - \widehat{\overline{\mathcal{H}}}_r(\beta)\right| \leq c_m B^m m^{\frac{m}{2}}\left(\frac{2\sqrt{2}mn}{Nb}L_f B_w^L\|X_\theta\|_F\left(1 + \sqrt{2L\ln(2)}\right) + 3\sqrt{\frac{\ln(4/\delta)}{2N}}\right).$$

**Remark 3.** *The results above imply the generalization errors $\varepsilon_1$ and $\varepsilon_3$ can converge at the rate of $1/\sqrt{N}$, considering for each row in $X_\theta$ its $L^2$-norm is bounded from above by $\sqrt{(m-1)\pi^2/4}$.*

*Obtaining the results require adaptations of classical Rademacher complexity techniques. The difficulty comes from the minimum form of $\rho(x, \theta) = \min_{i \in [m]}\left\{\frac{r_i - f_i(x)}{\lambda_i(\theta)}\right\}$, which differs from usual loss functions. We provide the detailed derivation in Appendix B.6. We further remark the scale assumption on $r_i - f_i(x)$ is mild. $f \circ x_{\beta^*}(\Theta)$ and $f \circ x_{\hat{\beta}}(\Theta)$ are supposed to approach $\mathcal{T}$, the Pareto Front, under proper optimization procedures; we are thus able to select a reference point $r$ far away enough from the set $\mathcal{T} \cup f \circ x_{\beta^*}(\Theta) \cup f \circ x_{\hat{\beta}}(\Theta)$ to make the assumption hold.*

## 6 Experiments

**Testing problems.** This section demonstrates that our method can generate high-quality continuous Pareto solutions by the Pareto neural model for multiobjective synthetic, design, and control

problems. We evaluated our approach's performance against established methods using well-known benchmark problems such as ZDT1-2 (m=2) [42], VLMOP1-2 (m=2) [38], and real-world design problems like Four Bar Truss Design (RE21, m=2), Hatch Cover Design (RE24, m=2), and Rocket Injector Design (m=3) [43] as well as MO-LQR (m=2,3) [44]. MO-LQRs serve as multiobjective reinforcement learning problems. For simplicity, except MO-LQR, all objectives are normalized to [0,1]. Problem details are described in Appendix A.3.

**Neural model architecture and feasibility guarantees.** We employed a 4-layer fully connected neural network, similar to [37], to construct our Pareto neural model $x_\beta(\cdot)$. The network is optimized using Stochastic Gradient Descent (SGD) with a batch size of 256 [4]. For constrained problems, to conform to the lower bound $(l)$ and upper bound $(u)$ constraints, a sigmoid activation function mapped the previous layer's output to these boundaries. For unconstrained problems, the output solution was derived from a linear combination of $x_{\mathrm{mid}}$. Further architecture details are available in Appendix A.2.

**Baseline methods.** Our approaches are compared to existing hypervolume-based multiobjective evolutionary algorithms (MOEAs) SMS-EMOA [14] and various Pareto set learning (PSL) methods. Before our approach was developed, three PSL approaches were developed, namely, EPO-based and Tchebycheff-based. Baseline methods are listed as follows,

1. One hypervolume-based multiobjective evolutionary algorithm: SMS-MOEA [14], which uses evolutionary algorithms to maximize the hypervolume of a finite set of solutions.
2. The LS-based PSL [12], optimizing $\min_\beta \mathbb{E}_{\lambda \sim \mathrm{Unif}(\Delta^{m-1})}[\sum_{i=1}^m \lambda_i f_i(x_\beta(\theta))]$, where $\Delta^{m-1}$ is the $(m-1)$-simplex. Since linear scalarization can only find the convex part of the Pareto front [39], theoretically, this method cannot recover the full Pareto front.
3. The EPO-based PSL [12], optimizing $\min_\beta \mathbb{E}_{\lambda \sim \mathrm{Unif}(\Delta^{m-1})}[\mathrm{EPO}(f(x_\beta(\theta)), \lambda)]$, the expectation of EPO loss [3]. The EPO-based PSL is slow since each solving for each EPO subproblem, which involves an expensive optimization problem.
4. The Tchebycheff-based PSL [23, 37], optimizing $\min_\beta \mathbb{E}_{\lambda \sim \mathrm{Unif}(\Delta^{m-1})}[\max_{i \in [m]} \lambda_i f_i(x_\beta(\theta))]$.

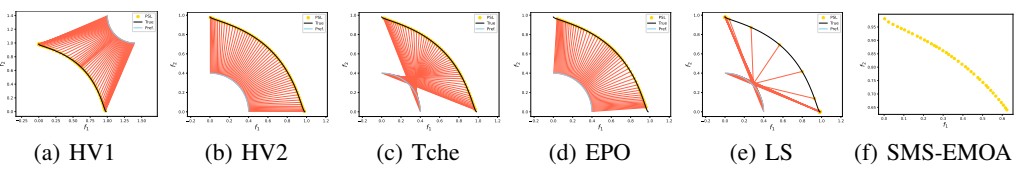

| (a) HV1 | (b) HV2 | (c) Tche | (d) EPO | (e) LS | (f) SMS-EMOA |

Figure 6: Result comparisons on VLMOP2.

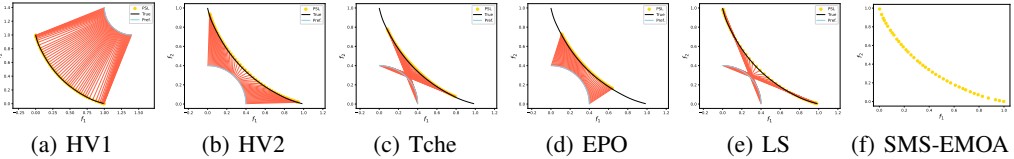

| (a) HV1 | (b) HV2 | (c) Tche | (d) EPO | (e) LS | (f) SMS-EMOA |

Figure 7: Result comparisons on the Four Bar Truss Design problem.

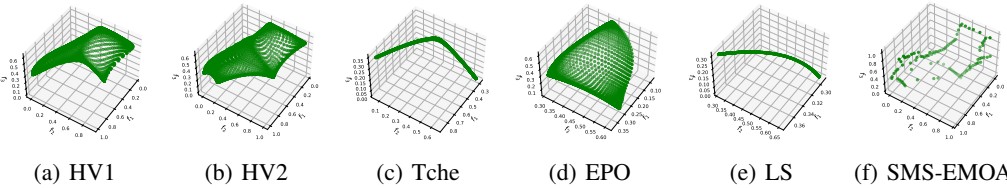

| (a) HV1 | (b) HV2 | (c) Tche | (d) EPO | (e) LS | (f) SMS-EMOA |

Figure 8: Result comparisons on the Rocket Injector Design problem.

**(Metrics.)** To evaluate our learned Pareto solutions, we use three indicators. These are: (1) the Hypervolume (HV), which measures the covered objective space, where a high value indicates greater

---

[4]Batch size is 128 for EPO-based PSL on LQRs since EPO is extremely slow.

Table 1: The training/prediction time of PSL-HV, and running time of SMS-EMOA in Rocket Injector Design.

| | SMS-EMOA | MOEA/D | Ours (Train) | Ours (Predict) |
|---|---|---|---|---|
| Time | 6.2m | 1.5m | $\approx 40$s | $<1$s |

coverage; (2) the `Range` indicator, which measures the extent of the Pareto front learned by the model, where a larger Range is preferred; and (3) the `Sparsity` metric, which evaluates the degree of sparsity in the learned Pareto front, where a lower sparsity value indicates a denser Pareto front. See Appendix A.1 for formal metric definitions.

Key findings from experiments are summarized as follows.

Table 2: PSL results on all problems. For PSL-HV1, $r = 0.6 + y^{\text{nadir}}$. For Tche-based PSL, the reference point $z = 1.1 \cdot y^{\text{ideal}}$, as claimed in their original paper.

| | ZDT1 | | | | ZDT2 | | | | VLMOP1 | | | |
|---|---|---|---|---|---|---|---|---|---|---|---|---|
| Method | HV↑ | Range↑ | Sparsity↓ | Time(s)↓ | HV | Range | Sparsity | Time(s) | HV | Range | Sparsity | Time(s) |
| PSL-EPO | 10.82 | 0.94 | 0.14 | 90.7 | 11.21 | 1.43 | 0.86 | 91.1 | 11.55 | 1.18 | 0.11 | 90.79 |
| PSL-LS | **11.89** | **1.56** | 1.07 | 9.17 | 8.75 | 0.0 | 0.0 | 9.03 | 12.05 | 1.56 | 0.5 | 9.15 |
| PSL-Tche | 11.49 | 1.37 | 0.33 | 21.11 | 11.19 | 1.42 | 0.66 | 21.65 | 11.83 | 1.42 | 0.22 | 21.61 |
| PSL-HV1 | 11.79 | 1.52 | 0.72 | 31.79 | 11.3 | 1.46 | 0.73 | 36.68 | **12.07** | **1.57** | 0.84 | 32.24 |
| PSL-HV2 | 11.77 | 1.52 | 0.7 | 25.71 | **11.51** | **1.55** | 1.0 | 25.8 | 12.02 | 1.54 | 0.46 | 25.45 |
| | VLMOP2 | | | | Four Bar Truss Design | | | | Hatch Cover Design | | | |
| PSL-EPO | 11.37 | 1.49 | 0.63 | 90.95 | 11.2 | 1.18 | 0.21 | 255.44 | 11.96 | 1.16 | 0.07 | 649.12 |
| PSL-LS | 11.51 | **1.57** | 7.22 | 9.57 | 11.89 | **1.56** | 0.94 | 41.75 | 12.14 | 1.5 | 0.27 | 113.52 |
| PSL-Tche | 11.47 | 1.52 | 0.63 | 22.14 | 11.5 | 1.37 | 0.31 | 55.37 | 12.06 | 1.37 | 0.08 | 125.43 |
| PSL-HV1 | **11.57** | **1.57** | 0.98 | 32.52 | **11.9** | **1.56** | 0.81 | 66.6 | **12.18** | **1.53** | 0.7 | 131.36 |
| PSL-HV2 | 11.56 | 1.56 | 0.66 | 25.78 | 11.82 | 1.53 | 0.65 | 60.03 | 12.15 | 1.51 | 0.37 | 130.4 |
| | MO-LQR2 | | | | Rocket Injector Design | | | | MO-LQR3 | | | |
| PSL-EPO | 3.42 | 0.3 | 0.23 | 546.81 | 37.23 | 0.68 | 0.81 | 133.03 | 2.41 | 0.31 | 2.33 | 1149.4 |
| PSL-LS | 3.59 | 0.5 | 0.84 | 296.98 | 35.55 | 0.19 | 0.88 | 21.35 | 2.58 | 0.37 | 1.9 | 408.59 |
| PSL-Tche | 3.52 | 0.39 | 0.44 | 314.75 | 37.73 | 0.19 | 1.34 | 34.01 | 2.54 | 0.38 | 2.19 | 424.15 |
| PSL-HV1 | **3.61** | **0.73** | 3.62 | 310.01 | 40.84 | **0.7** | 3.31 | 39.09 | **2.62** | **0.59** | 14.7 | 430.72 |
| PSL-HV2 | **3.61** | 0.59 | 2.19 | 312.95 | **40.94** | **0.7** | 5.54 | 36.63 | 2.6 | 0.55 | 8.93 | 424.51 |

① **(Comparison with SMS-EMOA.)** PSL and HV-based evolutionary algorithms serve as different purposes. PSL is designed to find an *infinite* set of solutions using a neural network model, while EMOAs aim to find a *finite* solution set. This difference becomes evident in three-objective problems, as shown in Figure 8, where 100 evenly distributed solutions by traditional MOEAs fall short of approximating the full Pareto set/front. PSL requires only several seconds of training time and can quickly predict highly approximated Pareto solutions for other unknown preferences. In contrast, SMS-EMOA typically takes a much longer time, especially for objectives larger than three. The advantage of evolution-based SMS-EMOA is its ability to find global optimal solutions, while current PSL models still struggle with handling local optimas.

Table 2 reports the mean results for various PSL approaches under five random seeds. The standard deviation values can be found in Appendix A.4. All experiments were conducted with 1000 iterations for PSL-HV1, PSL-HV2, LS-based PSL, and Tchebycheff-based PSL, while EPO-based PSL is limited to 100 iterations due to time limitation.

② **(Comparison with LS-based PSL.)** It is well-recognized that LS-based PSL [12] recovers only the convex part of a Pareto front [39], resulting in finding a small portion of Pareto solutions for concave problems. For example, in the case of ZDT2, LS-based PSL could only find two extreme points. We would like to mention another less-discussed limitation of linear scalarization: the non-uniform distribution of Pareto objectives under uniformly distributed preferences. This issue is evident in problems like ZDT1 and VLMOP2 where Pareto objectives are densely distributed at the front's margins, but fewer objectives are distributed around the center of the front.

③ **(Comparison with EPO-based PSL.)** As mentioned in Remark 1, both EPO and our proposed method (under mild conditions) can find the exact Pareto objective aligned with the preference vector. However, our proposed approach is approximately 40+ times faster than EPO-based PSL. Unlike the EPO-based PSL method, our approach only requires calculating the gradient of one objective function, avoiding the need to compute gradients for all objectives and solve complex optimization problems as described in their work [3]. This efficiency advantage is particularly evident in

MO-LQR problems, where EPO-based PSL takes significantly more time to execute. Furthermore, we have observed that EPO-based PSL does not perform well on tasks ZDT1 and Rocket Injector Design.

④ (**Balancing range and sparsity of a PF.**) When simply running PSL, we observed a tendency for the Pareto objectives to concentrate on the central portion of the Pareto front, neglecting solutions that correspond to boundary preferences, such as $f = [0,1]/[1,0]$ for bi-objective problems. To address this observation, a simple technique is to assign higher weights to the boundary objectives. For PSL-HV1, a larger reference point $r$ is suggested to be used in comparison to the nadir point, and therefore more coordinates will correspond to boundary Pareto objectives (refer to Remark 4 in Appendix B.4). However, setting the reference point to be excessively large can result in a sparse learned Pareto front; we empirically set this offset to be 0.6 (the empirical study is in Figure 10 and 11). In PSL-HV2, the preference $\lambda$ is placed in the denominator, resulting in greater weighting for the boundary Pareto objectives when the preference value approaches zero.

PSL-HV2 is related to the modified Tchebycheff (mTche) scalarization function [45], but the latter has different sampling techniques and lacks a hypervolume interpretation. When $\lambda$ approaches [0,1] or [1,0], both mTche and PSL-HV2 faces numerical challenges resulting in a large gradient, which can be effectively addressed the gradient clipping technique [46, 47] with a clipping norm of a certain value (e.g., 4.0 is used). Figure 9 shows that the learning process is unstable without gradient clipping but is stabilized when gradient clipping is applied. In contrast, PSL-HV1 involves minimization rather than maximization in its inner optimization problem and does not suffer from gradient explosion.

## 7 Conclusion, Limitation, and Future Work

This paper presented a method to learn the set of all Pareto solutions through hypervolume maximization. The method was motivated by a previous lemma, which allows for hypervolume estimation of infinite solutions. We further gave a precise geometric perspective on Pareto set learning; we studied the generalization gap between estimated and true hypervolumes, and discussed the key techniques to train a Pareto neural model that outperformed previous methods. The proposed method will also be incorporated into the `Moon` library [48].

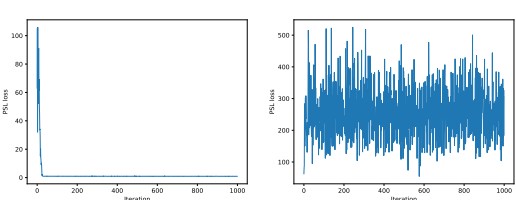

(a) With gradient clipping. (b) W.o. gradient clipping.

Figure 9: Learning curves of PSL-HV2 on VLMOP2.

**Limitation and future works.** The acknowledged constraint of our work is the reliance on gradient-based methods, which can result in locally optimal solutions when the objectives $f_i$'s are non-convex. To overcome this issue and provide more solid statistical guarantee, one feasible direction is to explore classical nonparametric techniques that can improve the robustness of the method. Another way is to use some evolutionary methods such as [49, 50, 51, 52] to skip local optimal solutions. Additionally, we plan to apply the more robust Pareto neural models to large-scale multi-objective problems, such as molecular design and deep reinforcement learning.

## Acknowledgement

This paper is supported by the Research Grants Council of the Hong Kong Special Administrative Region, China [GRF Project No: CityU 11215622 ] and by Natural Science Foundation of China [Project No: 62276223]. In the last, we sincerely appreciate the helpful insights and feedbacks provided by Liang Zhao;Dake Bu;Yingying Yu (CityU Hong Kong) and Dr. Jingda Deng (XJTU).

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

## A  Experiment Details

### A.1  Metrics

We assess the performance of a Pareto neural model, denoted as $x_\beta(\cdot)$, by examining its output with the input of $\widehat{\Theta}$. Here, $\widehat{\Theta}$ represents a set of $N$ uniformly discrete angles. The resulting output solution set is denoted as $A$ and can be expressed as $A = \{y^{(1)}, \ldots, y^{(N)}\}$, where $A = f \circ x_\beta(\widehat{\Theta})$. The three metrics are:

1. The `Hypervolume` indicator [29], which measures both the diversity and convergence of $A$;

2. The `Range` indicator, which measure the angular span of $A$;

3. The `Sparsity` indicator [53], which measures the average distances between neighborhood points.

**(The Hypervolume Indicator.)**  The hypervolume indicator [29] measuring $A$ is standard, which has been defined in the main paper,

$$\mathcal{H}_r(A) = \Lambda(\{q \mid \exists p \in A : p \preceq q \text{ and } q \preceq r\}),$$

where $r$ is a reference vector, $r \succeq y^{\text{nadir}}$. For bi-objective problems, the reference point $r$ is set to [3.5, 3.5], whereas for three-objective problems, the reference point is set to [3.5, 3.5, 3.5].

**(The Range Indicator.)**  The range indicator of a set of solutions $A : A = \{y^{(1)}, \ldots, y^{(N)}\}$ measures the angular span of $A$ in their polar coordinates. Let $(\rho^{(i)}, \theta^{(i)})$ be the polar coordinate of objective vectors $y^{(i)}$ with a reference point $r$. The relationship between the Cartesian and polar coordinates is,

$$\begin{cases} y_1 = r_1 - \rho \sin\theta_1 \sin\theta_2 \ldots \sin\theta_{m-1} \\ y_2 = r_2 - \rho \sin\theta_1 \sin\theta_2 \ldots \cos\theta_{m-1} \\ \ldots \\ y_m = r_m - \rho \cos\theta_1. \end{cases} \tag{10}$$

The `Range` indicator can be defined as the minimal angle span among all angles.

$$\texttt{Range}(A) = \min_{i \in [m-1]} \max_{1 \le u < v \le N} \left\{ \left| \theta_i^{(u)} - \theta_i^{(v)} \right| \right\}. \tag{11}$$

**(The Sparsity Indicator.)**  The sparsity indicator first introduced in [53]. A small sparsity indicator is preferred, implying that the learned objective set is dense.

$$\texttt{Sparsity}(A) = \frac{1}{N-1} \sum_{j=1}^{m} \sum_{i=1}^{N-1} \left( \tilde{y}_j^{(i)} - \tilde{y}_j^{(i+1)} \right)^2 \tag{12}$$

where $\tilde{y}_j^{(i)}$ is the i-th solution, and the j-th objective values in the sorted list by the non-dominating sorting algorithm [6]. For a better illustration, the unit of the `Sparsity` indicator is $10^{-3}$ for bi-objective problems and $10^{-7}$ for three objective problems.

### A.2  Neural Model Architecture and Feasibility Guarantees

We use a 4-layer fully connected neural network similar to [37] for the Pareto neural model $x_\beta(\cdot)$. We optimize the network using Stochastic Gradient Descent (SGD) optimizer with a batch size of 256. The first three layers are,

$$\begin{aligned} x_\beta(\cdot) : \theta &\to \text{Linear}(m-1, 64) \to \text{ReLU} \\ &\to \text{Linear}(64, 64) \to \text{ReLU} \\ &\to \text{Linear}(64, 64) \to \text{ReLU} \to x_{\text{mid}}. \end{aligned} \tag{13}$$

For constrained problems, to satisfy the constraint that the solution $x_\beta(\lambda)$ must fall within the lower bound $(l)$ and upper bound $(u)$, a sigmoid activation function is used to map the previous layer's output to these boundaries,

$$
\begin{aligned}
x_{\text{mid}} &\rightarrow \text{Linear}(64, n) \rightarrow \text{Sigmoid} \\
&\rightarrow \odot(u - l) + l \rightarrow \text{Output } x_\beta(\lambda).
\end{aligned}
\tag{14}
$$

For unconstrained problems, the output solution is obtained through a linear combination of $x_{\text{mid}}$,

$$
x_{\text{mid}} \rightarrow \text{Linear}(64, n) \rightarrow \text{Output } x_\beta(\lambda). \tag{15}
$$

### A.3 Benchmark Multiobjective Problems

**Standard Multiobjective Optimization (MOO) problems.** ZDT1-2 [42] and VLMOP1-2 [38] are widely recognized as standard multi-objective optimization (MOO) problems and are commonly employed in gradient-based MOO methods. ZDT1 exhibits a convex Pareto front described by $(y_2 = 1 - \sqrt{y_1}, \ 0 \leq y_1 \leq 1)$. ZDT2 presents a non-convex Pareto front defined by $(y_2 = 1 - y_1^2, \ 0 \leq y_1 \leq 1)$, and the LS-based PSL approach can only capture a small number of Pareto solutions.

**Real world designing problem.** Three real-world design problems with multi-objective optimization are the Four Bar Truss Design (RE21), Hatch Cover Design (RE24), and Rocket Injector Design (RE37). In order to simplify the optimization process, the objectives have been scaled to a range of zero to one.

**Multiobjective Linear Quadratic Regulator.** The Multiobjective Linear Quadratic Regulator (MO-LQR) problem is first introduced in [44]. MO-LQR is regarded as a specialized form of multi-objective reinforcement learning, where the problem is defined by a set of dynamics presented through the following equations:

$$
\begin{cases}
s_{t+1} = As_t + Ba_t \\
a_t \sim \mathcal{N}(K_{\text{LQR}} \cdot s_t, \Sigma).
\end{cases}
\tag{16}
$$

In accordance with the settings discussed in the aforementioned work by Parisi et al. [44], $A$, $B$, and $\Sigma$ are adopted as identity matrices. The initial state for the bi-objective problem is set to $s_0 = [10, 10]$, whereas for the three-objective problem, it is set to $s_0 = [10, 10, 10]$. The reward function is defined as $r_i(s_t, a_t)$, where $i$ represents the respective objective. The function is formulated as follows:

$$
r_i(s_t, a_t) = -(1 - \xi)(s_{t,i}^2 + \sum_{i \neq j} a_{t,i}^2) - \xi(a_{t,i}^2 + \sum_{i \neq j} s_{t,i}^2). \tag{17}
$$

Here, $\xi$ is the hyperparameter value that has been set to 0.1. The ultimate objective of the MO-LQR problem is to optimize the total reward while simultaneously taking into account the discount factor of $\gamma = 0.9$. The control matrix $K_{\text{LQR}}$ is assumed to be a diagonal matrix, and the diagonal elements of this matrix are treated as decision variables. The objectives are scaled with 0.01 for better illustration purposes.

Table 3 shows the number of decision variables and objectives used in this paper.

### A.4 Results on All Problems

Results for all the examined problems are depicted in Figures 12-20, and combined with the results tabulated in Table 2 of the main paper, several conclusions can be made.

**Behavior of LS-based PSL.** The linear scalarization function can only learn the convex part of a Pareto front [39]. This fact is validated by Figure 13(e), where LS-based PSL can only learn several discrete solutions.

Table 3: Problem information for multiobjective synthetic, design, and LQR problems.

| Problem | $m$ | $n$ |
|---|---|---|
| ZDT1 | 2 | 5 |
| ZDT2 | 2 | 5 |
| VLMOP1 | 2 | 5 |
| VLMOP2 | 2 | 5 |
| LQR2 | 2 | 2 |
| Four Bar Truss Design | 2 | 4 |
| Hatch Cover Design | 2 | 2 |
| Rocket Injector Design | 3 | 4 |
| LQR3 | 3 | 3 |

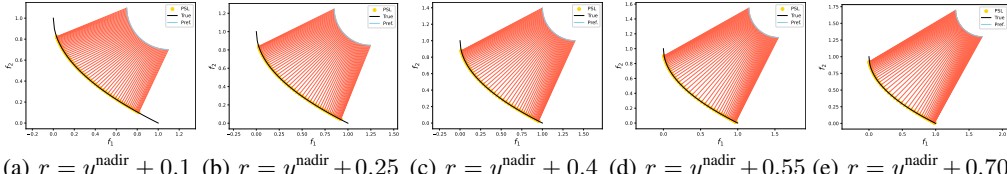

(a) $r = y^{\text{nadir}} + 0.1$ (b) $r = y^{\text{nadir}} + 0.25$ (c) $r = y^{\text{nadir}} + 0.4$ (d) $r = y^{\text{nadir}} + 0.55$ (e) $r = y^{\text{nadir}} + 0.70$

Figure 10: Hyperparameter study on ZDT1.

However, it is crucial to note that the connection between a solution and its corresponding preference vector, $\lambda(\theta)$, is *non-uniform*. This observation was rarely discussed in previous literature. Therefore, a uniform sampling of preferences will not result in a uniform sampling of Pareto objectives. This observation is supported by the results depicted in Figures 12(e), 15(e), and 17(e), where the learned Pareto objectives by LS-based PSL are not uniformly distributed. And as a result, the sparsity indicators are rather high, which indicates the learned front is sparse in some regions.

**Time Consumption of EPO-based PSL.** In comparison to our approach, the Exact Pareto Optimization [3] algorithm, which serves as the foundation for EPO-based PSL [12], exhibits low efficiency due to two factors.

1. To execute the Exact Pareto Optimization (EPO) algorithm, it is necessary to compute the gradients of all objectives, $\nabla f_i(x)$'s. This prerequisite entails performing $m$ back-propagations in pytorch, resulting in higher computational costs. In contrast, our approach relies just one back-propagation operation, rendering it a more efficient option in comparison to EPO.

2. For each iteration, the Exact Pareto Optimization (EPO) algorithm entails solving a complicated optimization problem based on the specific value of $f_i$'s, utilizing the respective gradients of $\nabla f_i(x)$'s. In contrast, our method does not rely on solving optimization problems for each iteration.

**Trade-off between Range and Sparsity.** We find that the reference point $r$ plays a crucial role in the performance of the final learned Pareto front for PSL-HV1. To demonstrate this effect, we consider different values of $r$ from the set $[0.1, 0.25, 0.4, 0.55, 0.7]$ in addition to the nadir point of each problem. The results are shown in the following two problems on the ZDT1 and ZDT2 problems.

Based on our empirical findings, emphasizing boundary Pareto objectives is crucial for recovering a complete Pareto set. From Figure 10 and 11, we observe that treating all coordinates $\theta$ equally (small nadir point) typically results in a partial recovery of the Pareto set. When $r$ exceeds a certain threshold (e.g., 0.4), raising its value directs Pareto objectives towards the boundary of the Pareto front, leading to poor uniformity of Pareto objective distribution.

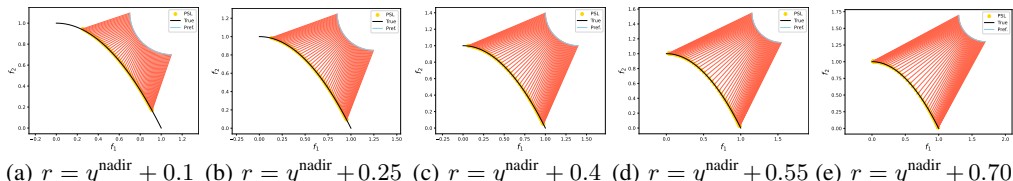

(a) $r = y^{\text{nadir}} + 0.1$ (b) $r = y^{\text{nadir}} + 0.25$ (c) $r = y^{\text{nadir}} + 0.4$ (d) $r = y^{\text{nadir}} + 0.55$ (e) $r = y^{\text{nadir}} + 0.70$

Figure 11: Hyperparameter study on ZDT2.

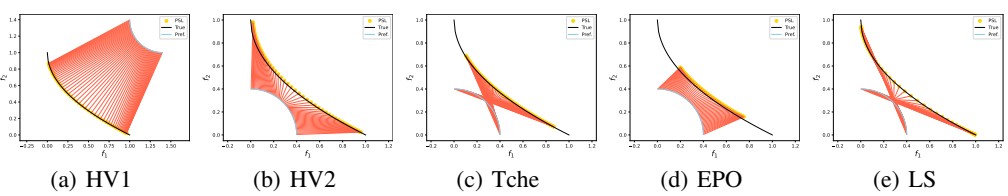

(a) HV1      (b) HV2      (c) Tche      (d) EPO      (e) LS

Figure 12: ZDT1.

## A.5 Pseudocode

For completeness, we provide the pseudocode of the proposed method. PSL-HV1 is selected as an example. $\eta$ is a positive learning rate.

---

**Algorithm 1: PSL-HV1.**

---

**Input:** The multiobjective function $f(\cdot)$ and initial Pareto neural model parameter $\beta$.

**for** $k=1$: $N_{iter}$ **do**

     Sample $N$ preference angles $\theta^{(1)}, \ldots, \theta^{(N)}$ from the positive unit sphere $\mathbb{S}_+^{m-1}$.

     Calculate the preference $\lambda^{(1)}, \ldots, \lambda^{(N)}$ by Equation (26).

     Calculate the empirical PSL gradient $\nabla_\beta \widehat{\overline{\mathcal{H}}}_r$ by Equation (28) with $N$ sampled preferences.

     Update $\beta$ by: $\beta \leftarrow \beta + \eta \nabla_\beta \widehat{\overline{\mathcal{H}}}_r$.

**end**

**Output:** The trained Pareto neural model $x_\beta(\cdot)$.

---

## B Characters of Hypervolume Maximization

### B.1 The Notation Table

To enhance the clarity of the paper, we have included a summary of the main notations in Table 4.

### B.2 Hypervolume Calculation in the Polar Coordinate

The idea of calculating the hypervolume is similar to the Lemma 5 proved in Appendix of [32]. The difference is that, we use a distance function $\rho_\mathcal{X}(\theta)$ as defined in Equation (6). For completeness, we provide the proof for Equation (5) in this section.

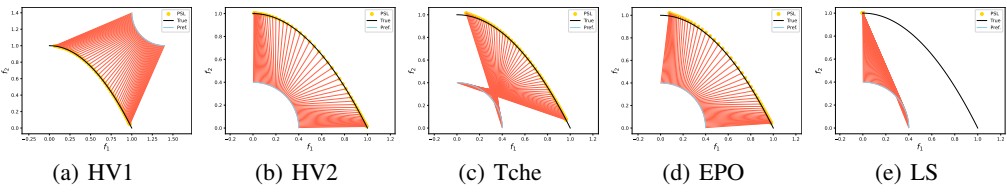

(a) HV1      (b) HV2      (c) Tche      (d) EPO      (e) LS

Figure 13: ZDT2.

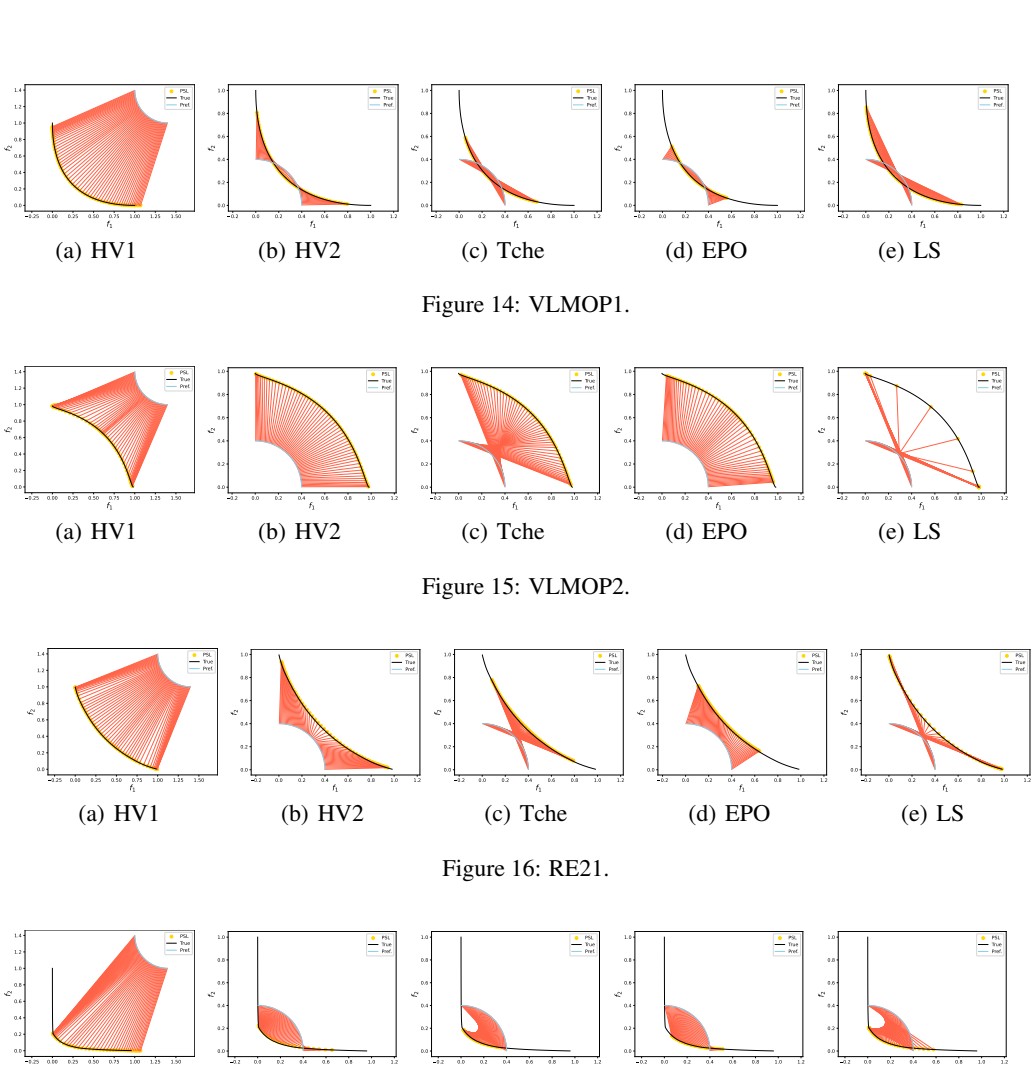

(a) HV1     (b) HV2     (c) Tche     (d) EPO     (e) LS

Figure 14: VLMOP1.

(a) HV1     (b) HV2     (c) Tche     (d) EPO     (e) LS

Figure 15: VLMOP2.

(a) HV1     (b) HV2     (c) Tche     (d) EPO     (e) LS

Figure 16: RE21.

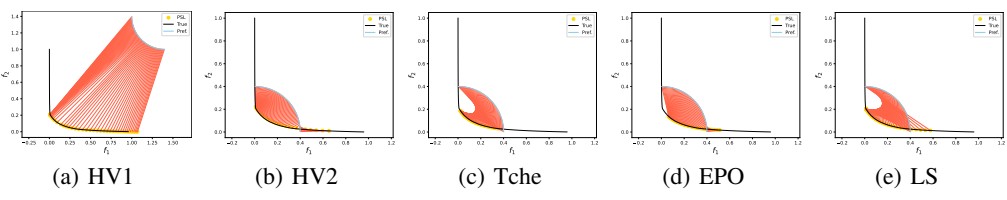

(a) HV1     (b) HV2     (c) Tche     (d) EPO     (e) LS

Figure 17: RE24.

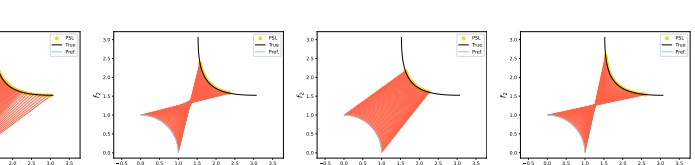

(a) HV1     (b) HV2     (c) Tche     (d) EPO     (e) LS

Figure 18: MO-LQR2.

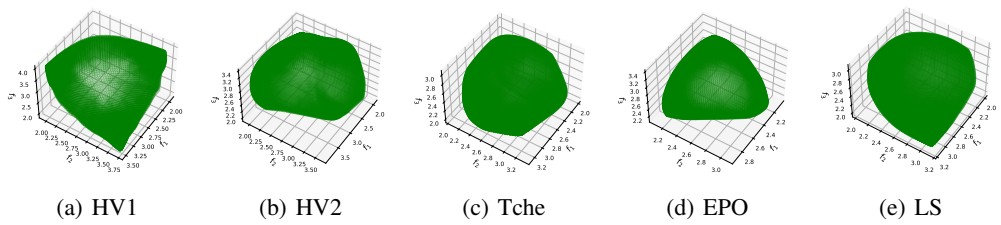

(a) HV1     (b) HV2     (c) Tche     (d) EPO     (e) LS

Figure 19: MO-LQR3.

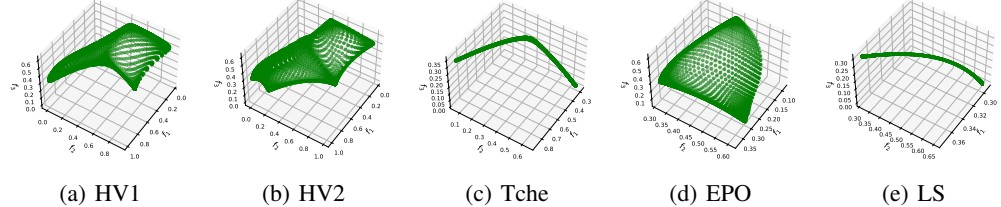

| (a) HV1 | (b) HV2 | (c) Tche | (d) EPO | (e) LS |

Figure 20: RE37.

Table 4: The notation table.

| Variable | Definition |
|---|---|
| $x$ | The decision variable. |
| $\mathcal{X}/\mathcal{Y}$ | The decision/objective space. |
| $n$ | The number of the decision variables. |
| $N$ | The number of samples. |
| $m$ | The number of objectives. |
| $\theta$ | The angular polar coordinate. |
| $\lambda(\theta), \lambda$ | An $m$-dimensional preference vector. |
| $\beta$ | The model parameter. |
| $y^{\text{nadir}}/y^{\text{ideal}}$ | The nadir/ideal point of a given MOO problem. |
| $\mathcal{T}$ | The Pareto front. |
| $\mathcal{H}_r(A)$ | The hypervolume of set $A$ w.r.t a reference . |
| $\mathbb{S}^{m-1}_+$ | The $(m\text{-}1)$-D positive unit sphere. |

*Proof.* $\mathcal{H}_r(\mathcal{T})$ can be simplified by the following equations. Here $\Omega$ denoted the dominated regions by the Pareto front, i.e., $\mathcal{H}_r(\mathcal{T}) = \Lambda(\Omega, r)$.

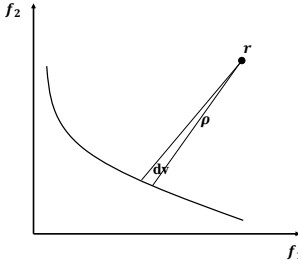

Figure 21: The hypervolume equals to the integral of dv.

$$\mathcal{H}_r(\mathcal{T}) = \int_{\mathbb{R}^m} I_\Omega dy_1 \ldots dy_m$$

("dv" denoted the infinitesimal sector area. )

$$= \underbrace{\int_0^{\frac{\pi}{2}} \ldots \int_0^{\frac{\pi}{2}}}_{m-1} dv$$

(dv equals the angle ratio multiplied by $\rho^m$ multiplied by the unit for volume. )

$$= \underbrace{\int_0^{\frac{\pi}{2}} \ldots \int_0^{\frac{\pi}{2}}}_{m-1} \overline{c}_m \cdot \frac{\rho_{\mathcal{X}}(\theta)^m}{2\pi \cdot \pi^{m-2}} \underbrace{d\theta_1 \ldots d\theta_{m-1}}_{d\theta} \tag{18}$$

(Re-arranging the terms.)

$$= \frac{\overline{c}_m}{2\pi^{m-1}} \underbrace{\int_0^{\frac{\pi}{2}} \ldots \int_0^{\frac{\pi}{2}}}_{m-1} \rho_{\mathcal{X}}(\theta)^m d\theta$$

$$= \frac{\overline{c}_m}{2\pi^{m-1}} \cdot \left(\frac{\pi}{2}\right)^{m-1} \cdot \mathbb{E}_\theta[\rho_{\mathcal{X}}(\theta)^m]$$

$$= c_m \mathbb{E}_\theta[\rho_{\mathcal{X}}(\theta)^m].$$

We specify $\theta \sim \text{Unif}(\Theta) = \text{Unif}([0, \frac{\pi}{2}]^{m-1})$ in Equation (18).

Line 2 holds since it represents the integral of $\Omega$ expressed in polar coordinates, wherein the element $dv$ corresponds to the volume associated with a segment obtained by varying $d\theta$.

Line 3 calculates the infinitesimal volume of $dv$ by noticing the fact that the ratio of $dv$ to $\overline{c}_m$ is $\frac{\rho_{\mathcal{X}}(\theta)^m}{2\pi \cdot \pi^{m-2}}$. Line 4 is a simplification of Line 3. And Line 5 and 6 express the integral in its expectation form. $\qquad\square$

### B.3 MOO Theory

In this section, we give the definitions and give proofs of the propositions used in the main paper.

We first formally define the term of aggregation function.

**Definition 2** (Aggregation function (modified from the Definition 2.6.1 in [54])). *A function* $g(\cdot, \lambda) : \mathbb{R}^m \mapsto \mathbb{R}$ *representing the preferences of the decision maker among the objective is called an aggregation function.* $g(\cdot, \lambda) : \mathbb{R}^m \mapsto \mathbb{R}$ *is decreasing, indicating that* $g(y^{(a)}, \lambda) < g(y^{(b)}, \lambda),$ *when* $y^{(a)} \prec y^{(b)}$.

This definition is slightly different from the original definition in Definition 2.6.1 in [54]), as we modify the condition from "strictly decreasing" to "decreasing". The considered aggregation function $\rho(x, \theta)$ does not satisfy the "strictly decreasing" condition in the original definition. When an aggregation function is only decreasing, the optimal objective can be weakly Pareto optimal. For this modified version of aggregation function, we have the following Lemma modified from Theorem 2.6.2 in [54].

**Lemma 2.** *Let* $y^*$ *belongs to the optimal solution set of* $g(\cdot, \lambda)$, *then* $y^*$ *is weakly Pareto optimal or Pareto optimal. Further, if* $y^*$ *be the only optimal solution of* $g(\cdot, \lambda)$, *then* $y^*$ *is Pareto optimal.*

We also provide the upper bound of the distance function $\rho_{\mathcal{X}}(\theta)$ in this section by the following lemma.

**Lemma 3** (The bound of in $\rho_{\mathcal{X}}(\theta)$ Equation (6)).

$$\rho_{\mathcal{X}}(\theta) \le Bm^{1/2}, \tag{19}$$

*when* $b \le r_i - f_i(x) \le B, \forall x \in \mathcal{X}, \forall i \in [m]$ *and* $\|\lambda(\theta)\| = 1$.

*Proof.* We show that the following inequalities hold,

$$
\begin{aligned}
\rho_{\mathcal{X}}(\theta) &\leq \max_{x \in \mathcal{X}, \|\lambda(\theta)\|=1} \left( \min_{i \in [m]} \left\{ \frac{r_i - f_i(x)}{\lambda_i(\theta)} \right\} \right) \\
&\quad (r_i - f_i(x) \leq B) \\
&\leq \max_{\|\lambda(\theta)\|=1} \left( \min_{i \in [m]} \left\{ \frac{B}{\lambda_i(\theta)} \right\} \right) \\
&\leq \frac{B}{m^{-1/2}} = Bm^{1/2}.
\end{aligned}
\tag{20}
$$

The transition from line one to line two is due to the fact that the inequality $r_i - f_i(x) \leq B$ holds for all $x \in \mathcal{X}$ and for all $i \in [m]$. The transition from line two to line three is $\max_{\|\lambda(\theta)\|=1} \left( \min_{i \in [m]} \left\{ \frac{B}{\lambda_i(\theta)} \right\} \right)$ is an optimization problem under the constraint $\|\lambda(\theta)\| = 1$. The upper bound for this optimization is when $\lambda_i = \ldots = \lambda_m = m^{-1/2}$.

$\square$

### B.4 $\rho_{\mathcal{X}}(\theta)$ as a Max-Min Problem

In this section, we first present the proof of Equation (6) from the main paper. To begin, we define the attainment surface $A_{\mathcal{T}}$ [33] to handle disjointed PFs. Intuitively, the attainment surface is continuous, containing the weakly Pareto objectives and Pareto objectives (see Figure 22).

**Definition 3** (Attainment surface [33]). *The attainment surface $A_{\mathcal{T}}$ induced by the Pareto front (PF) $\mathcal{T}$ is defined as the boundary set of the PF.*

$$
A_{\mathcal{T}} = \{ y \in \mathbb{R}^m | \exists y' \in \mathcal{T}, y' \preceq y, y' \nprec y \}.
\tag{21}
$$

With this definition, the distance $\rho_{\mathcal{X}}(\theta)$ between the reference point $r$ and the point coordinated at $\theta$ can be calculated using the following equation:

$$
\rho_{\mathcal{X}}(\theta) = \max_{x \in \mathcal{X}} \rho(x, \theta) = \max_{x \in \mathcal{X}} \min_{i \in [m]} \left\{ \frac{r_i - f_i(x)}{\lambda_i} \right\},
$$

where $\lambda_i$ are shorthand's for $\lambda_i(\theta)$.

*Proof.* We first define the point $P$.

**Definition 4.** *$P$ is defined as the intersection point of the ray from $r$ along angle $\theta$ and the attainment surface.*

There are two cases to consider:

① $P$ is a Pareto objective, i.e., $P \in \mathcal{T}$. ② $P$ belongs to the attainment surface but not to the Pareto front, i.e., $P \in (A_{\mathcal{T}} \setminus \mathcal{T})$.

Let $x^*$ be one of the optimal solutions of Problem $\max_{x \in \mathcal{X}} \rho(x, \theta)$. Next, we will separately demonstrate the validity of the above-mentioned statements.

① When $P$ is a Pareto objective, the according to Proposition 2, we have that $y = P = f(x^*)$ is the only objective of the problem $\tilde{g}(y, \lambda) = \max_{y \in \mathcal{Y}} \min_{i \in [m]} \left\{ \frac{r_i - y_i}{\lambda_i} \right\}$. And therefore, $\rho_{\mathcal{X}}(\theta) = \frac{r_i - f_i(x^*)}{\lambda_i}, \forall i \in [m]$.

② When $y = P \in (A_{\mathcal{T}} \setminus \mathcal{T})$. Then, according to Proposition 2, $y$ is one of the optimal objective of problem $\tilde{g}(y, \lambda) = \max_{y \in \mathcal{Y}} \min_{i \in [m]} \left\{ \frac{r_i - y_i}{\lambda_i} \right\}$. In such a case, according to the definition of attainment surface (Equation (21)), we have that there exist at least one index $j$, where $j = \arg\min \frac{r_j - f_j(x^*)}{\lambda_j}$ such that $\rho_{\mathcal{X}}(\theta) = \max_{x \in \mathcal{X}} \rho(x, \theta) = \max_{x \in \mathcal{X}} \min_{i \in [m]} \left\{ \frac{r_i - f_i(x)}{\lambda_i} \right\} = \min_{i \in [m]} \left\{ \frac{r_i - f_i(x^*)}{\lambda_i} \right\} = \frac{r_j - f_j(x^*)}{\lambda_j} = \frac{r_i - P_i}{\lambda_i}, i \in [m]$. $\square$

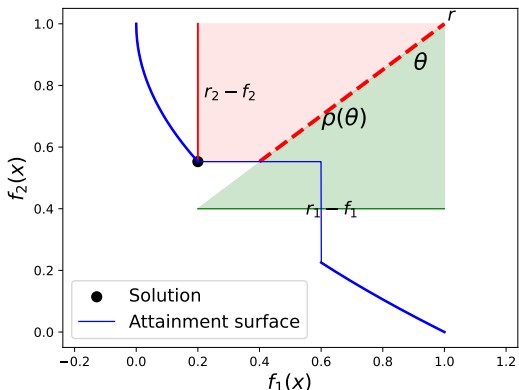

Figure 22: Illustration of the attainment surface on a case of a disjointed Pareto front.

The previous discussion explained that, in the case of a **disjointed Pareto front**, the distance between $r$ and the attainment surface corresponding to angle $\theta$ is denoted by $\rho_{\mathcal{X}}(\theta)$. In Figure 22, the black dot represents the solution for this scenario. It is worth noting that the integral of the distance function $\rho_{\mathcal{X}}(\theta)$ yields the hypervolume of a disjointed Pareto front, i.e., $\mathcal{H}_r(\mathcal{T}) = \mathcal{H}_r(A_{\mathcal{T}})$. This property aligns with the purpose of our paper.

In closing this section, we note that the absence of a (weakly) Pareto objective aligned with the preference angle $\theta$.

**Remark 4.** *When $P$ (in Definition 4) is not a (weakly) Pareto objective, The optimal solution $y^*$ w.r.t the problem $\tilde{g}(y, \lambda) = \max_{y \in \mathcal{Y}} \min_{i \in [m]} \left\{ \frac{r_i - y_i}{\lambda_i} \right\}$ belongs to the boundary of the Pareto front $\partial\mathcal{T}$. Further $y^* = \arg\min_{y \in \mathcal{Y}} \max_{i \in [m]} \left\{ \frac{y_i - P_i}{\lambda_i} \right\}$.*

### B.5 Pareto Front Hypervolume Calculation (Type2 and gradients)

In this section, we provide another form to calculate the Pareto front hypervolume. We first define region $A$ as the set of points dominating the Pareto front.

$$A = \{q \mid \exists p \in \mathcal{T} : p \preceq q \text{ and } q \succeq p^{\text{ideal}}\}. \tag{22}$$

We use the notation $\Lambda(\cdot)$ to represent the Lebesgue measure of a set. Geometrically, as illustrated in Figure 23, it can be observed that:

$$\Lambda(A) + \mathcal{H}_r(\beta) = \prod_{i=1}^{m}(r_i - y_i^{\text{ideal}}). \tag{23}$$

The volume of $A$ can be calculated in a polar coordinate as follows,

$$\Lambda(A) = c_m \int_{(0, \frac{\pi}{2})^{m-1}} \overline{\rho}_{\mathcal{X}}(\theta)^m d\theta, \tag{24}$$

Figure 23: The hypervolume calculation (Type2).

where $c_m$ is a constant and $\overline{\rho}_{\mathcal{X}}(\theta)$ represents the distance from the ideal point to the Pareto front at angle $\theta$. This distance function $\overline{\rho}_{\mathcal{X}}(\theta)$ is obtained by solving the optimization problem assuming that any ray from $\theta$ intersects with the Pareto front.

**Problem 1.**

$$\overline{\rho}_{\mathcal{X}}(\theta) = \min_{x \in \mathcal{X}} \overline{\rho}_{\mathcal{X}}(\theta, x) = \min_{x \in \mathcal{X}} \max_{i \in [m]} \left\{ \frac{f_i(x) - y_i^{ideal}}{\lambda_i(\theta)} \right\}, \quad \theta \in \left(0, \frac{\pi}{2}\right)^{m-1}. \tag{25}$$

The relationship between preference $\lambda$ and the polar angle $\theta$ is as follows:

$$\begin{cases} \lambda_1(\theta) = \sin\theta_1 \sin\theta_2 \ldots \sin\theta_{m-1} \\ \lambda_2(\theta) = \sin\theta_1 \sin\theta_2 \ldots \cos\theta_{m-1} \\ \ldots \\ \lambda_m(\theta) = \cos\theta_1. \end{cases} \tag{26}$$

Combining Equation (24) and Equation (25) implies that $\overline{\mathcal{H}}_r(\beta)$ can be estimated as an expectation problem,

$$\overline{\mathcal{H}}_r(\beta) = \prod_{i=1}^{m}(r_i - y_i^{\text{ideal}}) - \frac{1}{m}c_m \mathbb{E}_\theta[\overline{\rho}_{\mathcal{X}}(x_\beta(\theta), \theta)^m]. \tag{27}$$

**Remark 5.** *Compared to PSL-HV1, PSL-HV2 has weaker theoretical results due to its reliance on a stronger assumption. This assumption requires the presence of a (weakly) Pareto objective aligned on the ray originating from any coordinate $\theta$. An additional challenge with PSL-HV2 is that the Lipschitz constant cannot be bounded, making generalization theory for PSL-HV2 significantly more difficult (Equation (30)).*

We conclude this section by presenting the analytical expression for $\nabla_\beta \mathcal{H}_r(\beta)$. The gradient for PSL-HV1 can be computed using the chain rule, which yields:

$$\nabla_\beta \overline{\mathcal{H}}_r = \begin{cases} mc_m \mathbb{E}_\theta[\rho(x_\beta(\theta),\theta)^{m-1} \underbrace{\frac{\partial\rho(x_\beta(\theta),\theta)}{\partial x_\beta(\theta)}}_{1\times n} \underbrace{\frac{\partial x_\beta(\theta)}{\partial\beta}}_{n\times d}], & \rho(x_\beta(\theta),\theta) \geq 0. \\ c_m \mathbb{E}_\theta[\underbrace{\frac{\partial\rho(x_\beta(\theta),\theta)}{\partial x_\beta(\theta)}}_{1\times n} \underbrace{\frac{\partial x_\beta(\theta)}{\partial\beta}}_{n\times d}], & \text{Otherwise.} \end{cases} \tag{28}$$

where we specify $\theta$ to satisfy the law of $\text{Unif}(\Theta)$. The gradient of PSL-HV2 can be calculated by,

$$\nabla_\beta \overline{\mathcal{H}}_r = -mc_m \mathbb{E}_\theta[\overline{\rho}_{\mathcal{X}}(x_\beta(\theta),\theta)^{m-1}]\underbrace{\frac{\partial\overline{\rho}_{\mathcal{X}}(x_\beta(\theta),\theta)}{\partial x_\beta(\theta)}}_{1\times n} \underbrace{\frac{\partial x_\beta(\theta)}{\partial\beta}}_{n\times d}]. \tag{29}$$

## B.6 Proof of Theorem 1

**Definitions and preliminaries**. The proof will heavily utilize the existing results on Rademacher complexity of MLPs. We will first provide some useful definitions and facts. We start with the definition of Rademacher complexity as follows:

**Definition 5** (Rademacher complexity, Definition 13.1 in [55]). *Given a set of vectors $V \subseteq \mathbb{R}^n$, we define the (unnormalized) Rademacher complexity as*

$$\text{URad}(V) := \mathbb{E}\sup_{u\in V}\langle\epsilon, u\rangle,$$

*where each coordinate $\epsilon_i$ is an i.i.d. Rademacher random variable, meaning $\Pr[\epsilon_i = +1] = \frac{1}{2} = \Pr[\epsilon_i = -1]$. Furthermore, we can accordingly discuss the behavior of a function class $\mathcal{G}$ on $S = \{z_i\}_{i=1}^N$ by using the following set:*

$$\mathcal{G}_{|S} := \{(g(z_1), \ldots, g(z_N)) : g \in \mathcal{G}\} \subseteq \mathbb{R}^N,$$

*and its Rademacher complexity is*

$$\text{URad}(\mathcal{G}_{|S}) = \mathbb{E}_\epsilon \sup_{u\in\mathcal{G}_{|S}}\langle\epsilon, u\rangle = \mathbb{E}_\epsilon\sup_{g\in\mathcal{G}}\sum_i \epsilon_i g(z_i).$$

Utilizing Rademacher complexity, we can conveniently bound the generalization error via the following theorem:

**Theorem 2** (Uniform Generalization Error, Theorem 13.1 and Corollary 13.1 in [55]). *Let $\mathcal{G}$ be given with $g(z) \in [a,b]$ a.s. $\forall g \in \mathcal{G}$. We collect i.i.d. samples $S = \{z_i\}_{i=1}^{N}$ from the law of random variable $Z$. With probability $\geq 1 - \delta$,*

$$\sup_{g \in \mathcal{G}} \mathbb{E}g(Z) - \frac{1}{N}\sum_i g(z_i) \leq \frac{2}{N}\, \mathrm{URad}\left(\mathcal{G}_{|S}\right) + 3(b-a)\sqrt{\frac{\ln(2/\delta)}{2N}}.$$

Specifically, the Rademacher complexity in using MLP is provided by the following theorem:

**Theorem 3** (Rademacher complexity of MLP, Theorem 1 in [56]). *Let $1$-Lipschitz positive homogeneous activation $\sigma_i$ be given, and*

$$\mathcal{G}^{MLP} := \{\theta \mapsto \sigma_L\left(W_L\sigma_{L-1}\left(\cdots\sigma_1\left(W_1\theta\right)\cdots\right)\right) : \|W_i\|_{\mathrm{F}} \leq B_w\}$$

*Then*

$$\mathrm{URad}\left(\mathcal{G}^{MLP}_{|S}\right) \leq B_w^L \|X_\theta\|_F (1 + \sqrt{2L\ln(2)}).$$

We can then utilize the following composition character of Rademacher complexity, to help induce the final Rademacher complexity of hypervolume.

**Lemma 4** (Rademacher complexity of compositional function class, adapted from Lemma 13.3 in [55]). *Let $g : \Theta \to \mathbb{R}^n$ be a vector of $n$ multivariate functions $g^{(1)}, g^{(2)}, \ldots, g^{(n)}$, $\mathcal{G}$ denote the function class of $g$, and further $\mathcal{G}^{(j)}$ be the function class of $g^{(j)}, \forall j$. We have a "partially Lipschitz continuous" function $\ell(g(\theta), \theta)$ so that $|\ell(g_1(\theta), \theta) - \ell(g_2(\theta), \theta)| \leq L_\ell \|g_1(\theta) - g_2(\theta)\|$ for all $g_1, g_2 \in \mathcal{G}$ and a certain $L_\ell > 0$; the associated function class of $\ell$ is denoted as $\mathcal{G}^\ell$. We then have*

$$\mathrm{URad}\left(\mathcal{G}^\ell_{|S}\right) \leq \sqrt{2}L_\ell \sum_{j=1}^{n} \mathrm{URad}\left(\mathcal{G}^{(j)}_{|S}\right).$$

*Proof.* This proof extends Lemma 13.3 in [55] for vector-valued $g$ and "partially Lipschitz continuous" $\ell$. We first similarly have

$$\mathrm{URad}\left(\mathcal{G}^\ell_{|S}\right) = \mathbb{E}\sup_{g \in \mathcal{G}}\sum_i \epsilon_i \ell(g(\theta_i), \theta_i)$$

$$= \frac{1}{2}\mathbb{E}_{\epsilon_{2:N}}\sup_{f,h \in \mathcal{G}}\left(\ell(f(\theta_1), \theta_1) - \ell(h(\theta_1), \theta_1) + \sum_{i=2}^{N}\epsilon_i\left(\ell(f(\theta_i), \theta_i) + \ell(h(\theta_i), \theta_i)\right)\right)$$

$$\leq \frac{1}{2}\mathbb{E}_{\epsilon_{2:N}}\sup_{f,h \in \mathcal{G}}\left(L_\ell\|f(\theta_1) - h(\theta_1)\| + \sum_{i=2}^{N}\epsilon_i\left(\ell(f(\theta_i), \theta_i) + \ell(h(\theta_i), \theta_i)\right)\right)$$

$$\leq \frac{1}{2}\mathbb{E}_\epsilon \sup_{f^{(j)}, h^{(j)} \in \mathcal{G}^{(j)}}\left(L_\ell\sqrt{2}\left|\sum_{j=1}^{n}\epsilon_1^{(j)}(f^{(j)}(\theta_1) - h^{(j)}(\theta_1))\right| + \right.$$

$$\left. \sum_{i=2}^{N}\epsilon_i\left(\ell(f(\theta_i), \theta_i) + \ell(h(\theta_i), \theta_i)\right)\right),$$

where $\epsilon_1^{(j)}$'s are new i.i.d. Rademacher variables; the last inequality comes from Proposition 6 in [57] (see Equations (5)-(10) in [57] for more details). We can then get rid of the absolute value by

considering swapping $f$ and $h$,

$$\sup_{f^{(j)}, h^{(j)} \in \mathcal{G}^{(j)}} \left( \sqrt{2} L_\ell \left| \sum_{j=1}^{n} \epsilon_1^{(j)} (f^{(j)}(\theta_1) - h^{(j)}(\theta_1)) \right| + \sum_{i=2}^{N} \epsilon_i \left( \ell(f(\theta_i), \theta_i) + \ell(h(\theta_i), \theta_i) \right) \right)$$

$$= \max \left\{ \sup_{f^{(j)}, h^{(j)} \in \mathcal{G}^{(j)}} \left( \sqrt{2} L_\ell \sum_{j=1}^{n} \epsilon_1^{(j)} (f^{(j)}(\theta_1) - h^{(j)}(\theta_1)) + \sum_{i=2}^{N} \epsilon_i \left( \ell(f(\theta_i), \theta_i) + \ell(h(\theta_i), \theta_i) \right) \right), \right.$$

$$\left. \sup_{f^{(j)}, h^{(j)} \in \mathcal{G}^{(j)}} \left( \sqrt{2} L_\ell \sum_{j=1}^{n} \epsilon_1^{(j)} (h^{(j)}(\theta_1) - f^{(j)}(\theta_1)) + \sum_{i=2}^{N} \epsilon_i \left( \ell(f(\theta_i), \theta_i) + \ell(h(\theta_i), \theta_i) \right) \right) \right\}$$

$$= \sup_{f^{(j)}, h^{(j)} \in \mathcal{G}^{(j)}} \left( \sqrt{2} L_\ell \sum_{j=1}^{n} \epsilon_1^{(j)} (f^{(j)}(\theta_1) - h^{(j)}(\theta_1)) + \sum_{i=2}^{N} \epsilon_i \left( \ell(f(\theta_i), \theta_i) + \ell(h(\theta_i), \theta_i) \right) \right).$$

We can thus upper bounded $\text{URad}\left(\mathcal{G}_{|S}^\ell\right)$ by

$$\frac{1}{2} \mathbb{E}_\epsilon \sup_{f^{(j)}, h^{(j)} \in \mathcal{G}^{(j)}} \left( \sqrt{2} L_\ell \sum_{j=1}^{n} \epsilon_1^{(j)} (f^{(j)}(\theta_1) - h^{(j)}(\theta_1)) + \sum_{i=2}^{N} \epsilon_i \left( \ell(f(\theta_i), \theta_i) + \ell(h(\theta_i), \theta_i) \right) \right)$$

$$= \mathbb{E}_\epsilon \sup_{g^{(j)} \in \mathcal{G}^{(j)}} \left( \sqrt{2} L_\ell \sum_{j=1}^{n} \epsilon_1^{(j)} g^{(j)}(\theta_1) + \sum_{i=2}^{N} \epsilon_i \ell(g(\theta_i), \theta_i) \right),$$

Repeating this procedure for the other coordinates, we can further have

$$\text{URad}\left(\mathcal{G}_{|S}^\ell\right) \le \sqrt{2} L_\ell \mathbb{E}_\epsilon \sup_{g^{(j)} \in \mathcal{G}^{(j)}} \left( \sum_{i=1}^{N} \sum_{j=1}^{n} \epsilon_i^{(j)} g^{(j)}(\theta_i) \right) \le \sqrt{2} L_\ell \sum_{j=1}^{n} \mathbb{E}_\epsilon \sup_{g^{(j)} \in \mathcal{G}^{(j)}} \left( \sum_{i=1}^{N} \epsilon_i^{(j)} g^{(j)}(\theta_i) \right),$$

which leads to our claim in the lemma. $\qquad\square$

**Proof of Theorem 1.** We are now geared up for the complete proof.

*Proof.* We first introduce the sketch of the proof. We mainly utilize Theorem 2 to attain the claimed results in Theorem 1. Specifically, we set the random sample set $S = \{\theta_i\}_{i=1}^{N}$, the function class $\mathcal{G}$ as $\{\theta \mapsto c_m \rho(x_\beta(\theta), \theta)^m\}$ (the assumption $r_i - f_i(x) \in [b, B]$ indicates that $\rho(x, \theta) = \min_{i \in [m]} \{ \frac{r_i - f_i(x)}{\lambda_i(\theta)} \} \ge b \ge 0$ and by the definition in Equation (7), $\rho_\beta(\theta)$ is thus always $\rho(x(\theta), \theta)^m$; $x_\beta(\cdot)$ is an $L$-layer MLP to be specified later). Applying Theorem 2, we can obtain that with probability at least $1 - \frac{\delta}{2}$,

$$\sup_{g \in \mathcal{G}} \mathbb{E}_\theta g(\theta) - \frac{1}{N} \sum_i g(\theta_i) \le \frac{2}{N} \text{URad}\left(\mathcal{G}_{|S}\right) + 3 c_m (B\sqrt{m})^m \sqrt{\frac{\ln(4/\delta)}{2N}},$$

where the definition of URad and $\mathcal{G}_{|S}$ can be found in Definition 5. Simply replacing $\mathcal{G}$ with $-\mathcal{G} := \{-g : g \in \mathcal{G}\}$, we can have the inequality of the other direction with probability at least $1 - \frac{\delta}{2}$:

$$\sup_{g \in -\mathcal{G}} \mathbb{E}_\theta g(\theta) - \frac{1}{N} \sum_i g(\theta_i) \le \frac{2}{N} \text{URad}\left(-\mathcal{G}_{|S}\right) + 3 c_m (B\sqrt{m})^m \sqrt{\frac{\ln(4/\delta)}{2N}}$$

$$\Rightarrow \sup_{g \in \mathcal{G}} \mathbb{E}_\theta - g(\theta) - \frac{1}{N} \sum_i -g(\theta_i) \le \frac{2}{N} \text{URad}\left(-\mathcal{G}_{|S}\right) + 3 c_m (B\sqrt{m})^m \sqrt{\frac{\ln(4/\delta)}{2N}}$$

$$\Rightarrow \sup_{g \in \mathcal{G}} \frac{1}{N} \sum_i g(\theta_i) - \mathbb{E}_\theta g(\theta) \le \frac{2}{N} \text{URad}\left(\mathcal{G}_{|S}\right) + 3 c_m (B\sqrt{m})^m \sqrt{\frac{\ln(4/\delta)}{2N}},$$

where we apply the property $\text{URad}\left(-\mathcal{G}_{|S}\right) = \text{URad}\left(\mathcal{G}_{|S}\right)$. We thus, with probability at least $1 - \delta$ (as a result of union bound), can upper bound $\sup_{g \in \mathcal{G}} \left| \mathbb{E}_\theta g(\theta) - \frac{1}{N} \sum_i g\left(\theta_i\right) \right|$ by

$$
\max \left\{ \sup_{g \in \mathcal{G}} \mathbb{E}_\theta g(\theta) - \frac{1}{N} \sum_i g\left(\theta_i\right), \sup_{g \in \mathcal{G}} \frac{1}{N} \sum_i g\left(\theta_i\right) - \mathbb{E}_\theta g(\theta) \right\}
$$

$$
\leq \frac{2}{N} \text{URad}\left(\mathcal{G}_{|S}\right) + 3 c_m (B\sqrt{m})^m \sqrt{\frac{\ln(4/\delta)}{2N}}.
$$

For the next step, we will upper bound $\text{URad}\left(\mathcal{G}_{|S}\right)$ by analyzing the structure of $c_m \rho(x_\beta(\theta), \theta)^m$ and utilizing the existing bound (see Theorem 3) for Rademacher complexity of MLP $x_\beta$.

The main idea of controlling $\text{URad}\left(\mathcal{G}_{|S}\right)$ is to obtain the "partially Lipschitz continuity" that $|\rho(x_\beta(\theta), \theta) - \rho(x_{\beta'}(\theta), \theta)| \leq L_\rho \|x_\beta(\theta) - x_{\beta'}(\theta)\|$ for a certain $L_\rho > 0$; with the "partially Lipschitz continuity" we can apply Lemma 4 and obtain the desired bound. For simplicity, we denote $x_\beta(\theta), x_{\beta'}(\theta)$ respectively as $x, x'$, and use $\lambda_j$'s as shorthand for $\lambda_j(\theta)$'s. We now expand the difference $|\rho(x_\beta(\theta), \theta) - \rho(x_{\beta'}(\theta), \theta)|$ as:

$$
\left| \min_{j \in [m]} \frac{r_j - f_j(x)}{\lambda_j} - \min_{k \in [m]} \frac{r_k - f_k(x')}{\lambda_k} \right|
$$

$$
= \max \left\{ \min_{j \in [m]} \frac{r_j - f_j(x)}{\lambda_j} - \min_{k \in [m]} \frac{r_k - f_k(x')}{\lambda_k}, \min_{k \in [m]} \frac{r_k - f_k(x')}{\lambda_k} - \min_{j \in [m]} \frac{r_j - f_j(x)}{\lambda_j} \right\}.
$$

If we respectively denote the minima index of the two finite-term minimization as $j^*$ and $k^*$, we can then upper bound $|\rho(x_\beta(\theta), \theta) - \rho(x_{\beta'}(\theta), \theta)|$ by

$$
\max \left\{ \frac{r_{k^*} - f_{k^*}(x)}{\lambda_{k^*}} - \frac{r_{k^*} - f_{k^*}(x')}{\lambda_{k^*}}, \frac{r_{j^*} - f_{j^*}(x')}{\lambda_{j^*}} - \frac{r_{j^*} - f_{j^*}(x)}{\lambda_{j^*}} \right\}
$$

$$
= \max \left\{ \frac{f_{k^*}(x') - f_{k^*}(x)}{\lambda_{k^*}}, \frac{f_{j^*}(x) - f_{j^*}(x')}{\lambda_{j^*}} \right\} \leq \max_{j \in \{j^*, k^*\}} \frac{|f_j(x) - f_j(x')|}{\lambda_j}
$$

$$
\leq \max_{j \in \{j^*, k^*\}} \frac{L_f |x - x'|}{\lambda_j}.
$$

We note there is a special property for $\lambda_j$ when $j$ is the minima index: as $\|\lambda\| = 1$, there must be a certain $\lambda_j \geq 1/\sqrt{m}$, and since $b \leq r_j - f_j(x) \leq B, \forall j$, we have

$$
\frac{b}{\lambda_{j^*}} \leq \frac{r_{j^*} - f_{j^*}(x')}{\lambda_{j^*}} \leq \frac{B}{1/\sqrt{m}} \Rightarrow \lambda_{j^*} \geq \frac{b}{\sqrt{m}B}. \tag{30}
$$

With this special property, we obtain

$$
|\rho(x_\beta(\theta), \theta) - \rho(x_{\beta'}(\theta), \theta)| \leq \frac{\sqrt{m}B}{b} L_f |x - x'|.
$$

We further have

$$
|c_m \rho(x_\beta(\theta), \theta)^m - c_m \rho(x_{\beta'}(\theta), \theta)^m|
$$

$$
= c_m |\rho(x_\beta(\theta), \theta) - \rho(x_{\beta'}(\theta), \theta)| \left( \sum_{k=1}^m \rho(x_\beta(\theta), \theta)^{m-k} \rho(x_{\beta'}(\theta), \theta)^{k-1} \right)
$$

$$
\leq c_m \frac{\sqrt{m}B}{b} L_f |x - x'| \, m (B\sqrt{m})^{m-1} = c_m \frac{m}{b} (B\sqrt{m})^m L_f |x - x'|,
$$

which establishes the "partially Lipschitz continuity". The distance function $\rho(x_\beta(\theta), \theta)$ is bounded by Lemma 3. We can then apply Lemma 4 and have

$$
\text{URad}\left(\mathcal{G}_{|S}\right) \leq \sqrt{2} c_m \frac{m}{b} (B\sqrt{m})^m L_f n \, \text{URad}\left(\mathcal{G}_{|S}^{\text{MLP}}\right)
$$

$$
\leq \sqrt{2} c_m \frac{m}{b} (B\sqrt{m})^m L_f n \cdot B_w^L \|X_\theta\|_F (1 + \sqrt{2L \ln(2)}).
$$

Combining the pieces above, we finally have

$$\sup_{g \in \mathcal{G}} \left| \mathbb{E}_\theta g(\theta) - \frac{1}{N} \sum_i g(\theta_i) \right|$$

$$\leq \frac{2}{N} \operatorname{URad}\left(\mathcal{G}_{|S}\right) + 3c_m(B\sqrt{m})^m \sqrt{\frac{\ln(4/\delta)}{2N}}$$

$$\leq c_m(B\sqrt{m})^m \left( \frac{2\sqrt{2}mn}{Nb} L_f \cdot B_w^L \|X_\theta\|_F (1 + \sqrt{2L \ln(2)}) + 3\sqrt{\frac{\ln(4/\delta)}{2N}} \right),$$

which is the generalization error bound we claim. $\qquad\square$

