# Appendix—Hypervolume Maximization: A Geometric View of Pareto Set Learning

## 1 A   Experiment Details

### 2 A.1   Metrics

3 As outlined in the main body of the paper, we utilize three metrics to evaluate the effectiveness of
4 the learned solutions. In particular, we assess the performance of a Pareto neural model $x_\beta(\cdot)$ by
5 examining the output of the model for $N$ angles that are uniformly distributed. The output solution
6 set $A = \{y^{(1)}, \ldots, y^{(N)}\}$, where $A = f \circ x_\beta(\widehat{\Theta})$. The three metrics are:

7     1. The *Hypervolume* indicator [30], which measures both the diversity and convergence of $A$;

8     2. The *Range* indicator, which measure the angular span of $A$;

9     3. The *Sparsity* indicator [4], which measures the distances between adjacent points.

### 10 A.1.1   The Hypervolume Indicator

11 The hypervolume indicator [30] used to measure $A$ is standard, which has been defined in the main
12 paper,

$$\mathcal{H}_r(A) = \Lambda(\{q \mid \exists p \in A : p \preceq q \text{ and } q \preceq r\}), \tag{11}$$

13 and $r$ is a reference vector, $r \succeq y^{\text{nadir}}$. For bi-objective problems, the reference point $r$ is set to [3.5,
14 3.5], whereas for three-objective problems, the reference point is set to [3.5, 3.5, 3.5].

### 15 A.1.2   The Range Indicator

16 The range indicator of a Pareto front is defined in polar coordinates and determines the angular span
17 of the front. Let $(\rho^{(i)}, \theta^{(i)})$ be the polar coordinate of objective vectors $y^{(i)}$ with a reference point $r$.
18 The relationship between of the Cartesian and polar coordinate is,

$$\begin{cases} y_1 = r_1 - \rho \sin\theta_1 \sin\theta_2 ... \sin\theta_{m-1} \\ y_2 = r_2 - \rho \sin\theta_1 \sin\theta_2 ... \cos\theta_{m-1} \\ ... \\ y_m = r_m - \rho \cos\theta_1. \end{cases} \tag{12}$$

19 Then, the *Range* indicator is defined as,

$$\text{Range}(A) = \min_{i \in [m]} \max_{\substack{u \in [N], v \in [N], \\ u \neq v}} \left\{ |\theta_i^{(u)} - \theta_i^{(v)}| \right\}. \tag{13}$$

20 The *Range* indicator can be defined as the minimum angle span across all angles.

Submitted to 37th Conference on Neural Information Processing Systems (NeurIPS 2023). Do not distribute.

### A.1.3 The Sparsity Indicator

The sparsity indicator first introduced in [4] measures how dense a set of solutions is. Small inter-solution distances result in a small sparsity indicator indicating a dense Pareto front can be found by the Pareto neural model. We make a modification for $m = 2$ since we find that the maximization operator is much more stable.

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

Table 4: Licences.

| Resource | Link | License |
|---|---|---|
| EPO | https://github.com/dbmptr/EPOSearch.git | MIT license |
| pymoo | https://pymoo.org/ | Apache License 2.0 |
| reproblems | https://ryojitanabe.github.io/reproblems/ | None |

gradients of $\nabla f_i(x)$'s. In contrast, our method does not rely on solving optimization problems for each iteration.

**Emphasis on Boundary Solutions.** Based on our empirical findings, it is crucial to put emphasis on boundary solutions when aiming to recover a complete Pareto set. As shown in Figure 12 and 14, if all coordinate $\theta$ are dealt with equally important, the neural model can only recover a partial part of the Pareto set. PSL-HV1 and PSL-HV2 have different behaviors on the three-objective Rocket Injector Design problem, as shown in Figure 18. PSL-HV2 algorithm has a tendency to accurately identify the complete boundary of the Pareto front, but it often overlooks intermediate solutions. In contrast, although PSL-HV1 method may not always recover the complete boundary, it generates a denser Pareto front.

## A.5 Licences

In this paper, we utilized various licenses, which are outlined in Table 4. All methods were implemented using Python and the PyTorch framework, with the SMS-EMOA algorithm being aggregated in pymoo.

Table 5: Standard derivation (std) value of PSL results on all problems.

| Method | ZDT1 | | | | ZDT2 | | | | VLMOP1 | | | |
|---|---|---|---|---|---|---|---|---|---|---|---|---|
| | HV↑ | Range↑ | Sparsity↓ | Time(s)↓ | HV | Range | Sparsity | Time(s) | HV | Range | Sparsity | Time(s) |
| PSL-EPO | 0.05 | 0.04 | 0.08 | 2.03 | 0.13 | 0.06 | 0.25 | 0.91 | 0.01 | 0.01 | 0.02 | 0.56 |
| PSL-LS | 0.0 | 0.0 | 0.2 | 0.43 | 0.0 | 0.0 | 0.0 | 0.36 | 0.0 | 0.0 | 0.05 | 0.76 |
| PSL-Tche | 0.01 | 0.0 | 0.01 | 0.56 | 0.01 | 0.0 | 0.22 | 0.79 | 0.01 | 0.01 | 0.02 | 0.54 |
| PSL-HV1 | 0.01 | 0.0 | 0.05 | 0.22 | 0.03 | 0.01 | 0.04 | 0.2 | 0.0 | 0.0 | 0.03 | 0.48 |
| PSL-HV2 | 0.01 | 0.0 | 0.04 | 0.29 | 0.01 | 0.0 | 0.21 | 0.95 | 0.01 | 0.0 | 0.04 | 1.15 |
| | VLMOP2 | | | | Four Bar Truss Design | | | | Hatch Cover Design | | | |
| PSL-EPO | 0.08 | 0.04 | 0.19 | 0.48 | 0.02 | 0.01 | 0.01 | 1.53 | 0.0 | 0.02 | 0.06 | 4.96 |
| PSL-LS | 0.03 | 0.01 | 8.69 | 0.06 | 0.0 | 0.0 | 0.08 | 0.12 | 0.0 | 0.0 | 0.31 | 1.21 |
| PSL-Tche | 0.01 | 0.0 | 0.04 | 0.49 | 0.02 | 0.01 | 0.02 | 1.71 | 0.0 | 0.01 | 0.02 | 2.99 |
| PSL-HV1 | 0.0 | 0.0 | 0.19 | 1.32 | 0.01 | 0.0 | 0.03 | 0.38 | 0.02 | 0.02 | 1.41 | 1.18 |
| PSL-HV2 | 0.01 | 0.0 | 0.13 | 0.15 | 0.0 | 0.0 | 0.01 | 1.79 | 0.0 | 0.0 | 0.11 | 1.42 |
| | LQR2 | | | | Rocket Injector Design | | | | LQR3 | | | |
| PSL-EPO | 0.01 | 0.01 | 0.03 | 15.46 | 1.34 | 0.08 | 0.1 | 1.12 | 0.01 | 0.02 | 0.71 | 24.21 |
| PSL-LS | 0.0 | 0.0 | 0.08 | 3.7 | 0.0 | 0.0 | 0.02 | 0.11 | 0.0 | 0.01 | 0.05 | 5.79 |
| PSL-Tche | 0.01 | 0.01 | 0.1 | 4.63 | 0.01 | 0.0 | 0.02 | 1.17 | 0.01 | 0.01 | 0.27 | 8.9 |
| PSL-HV1 | 0.0 | 0.0 | 0.22 | 1.83 | 0.09 | 0.01 | 0.18 | 0.14 | 0.0 | 0.02 | 0.68 | 1.34 |
| PSL-HV2 | 0.0 | 0.0 | 0.13 | 9.86 | 0.03 | 0.01 | 1.53 | 1.31 | 0.0 | 0.01 | 0.72 | 11.95 |

# B  Characters of Hypervolume Maximization

## B.1  The Notation Table

To enhance the clarity of the paper, we have included a summary of the main notations in Table 6.

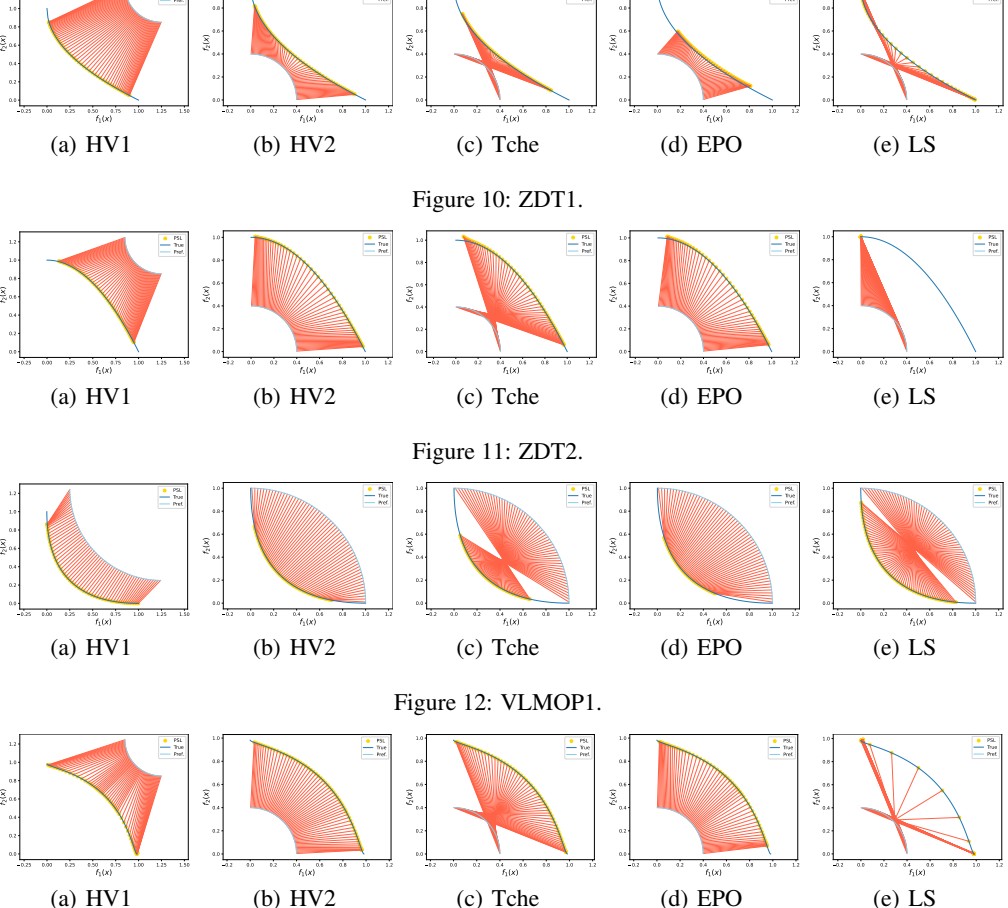

Figure 10: ZDT1.

(a) HV1  (b) HV2  (c) Tche  (d) EPO  (e) LS

Figure 11: ZDT2.

(a) HV1  (b) HV2  (c) Tche  (d) EPO  (e) LS

Figure 12: VLMOP1.

(a) HV1  (b) HV2  (c) Tche  (d) EPO  (e) LS

Figure 13: VLMOP2.

## B.2  Hypervolume Calculation in the Polar Coordinate

*Proof.* In this subsection, we provide the proof for Equation (5). $\mathcal{H}_r(\mathcal{F}^*)$ can be simplified by the following equations,

$$
\begin{aligned}
\mathcal{H}_r(\mathcal{F}^*) &= \int_{\mathbb{R}^m} I_\Omega dy_1 \ldots dy_m \\
&= \underbrace{\int_0^{\frac{\pi}{2}} \ldots \int_0^{\frac{\pi}{2}}}_{m-1} dv \\
&= \underbrace{\int_0^{\frac{\pi}{2}} \ldots \int_0^{\frac{\pi}{2}}}_{m-1} \overline{c}_m \cdot \frac{\rho_\mathcal{X}(\theta)^m}{2\pi \cdot \pi^{m-2}} \underbrace{d\theta_1 \ldots d\theta_{m-1}}_{d\theta} \\
&= \frac{\overline{c}_m}{2\pi^{m-1}} \underbrace{\int_0^{\frac{\pi}{2}} \ldots \int_0^{\frac{\pi}{2}}}_{m-1} \rho_\mathcal{X}(\theta)^m d\theta \\
&= \frac{\overline{c}_m}{2\pi^{m-1}} \cdot \left(\frac{\pi}{2}\right)^{m-1} \cdot \mathbb{E}_{\theta \sim \text{Unif}(\Theta)}[\rho_\mathcal{X}(\theta)^m] \\
&= c_m \mathbb{E}_{\theta \sim \text{Unif}(\Theta)}[\rho_\mathcal{X}(\theta)^m].
\end{aligned}
\tag{20}
$$

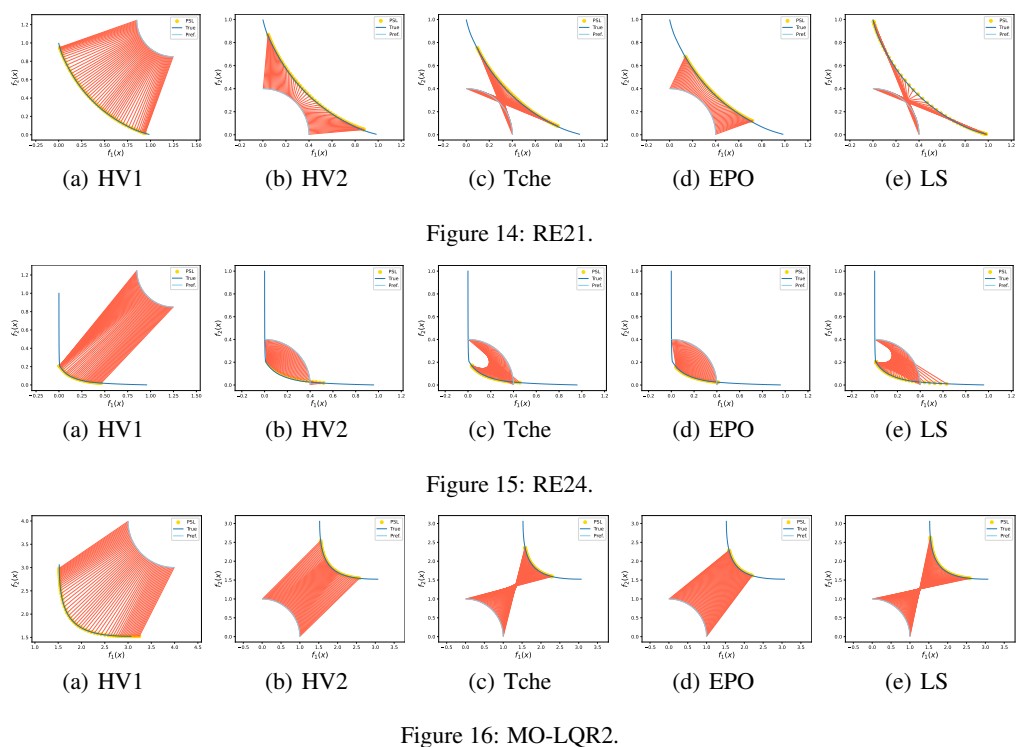

Figure 14: RE21.

Figure 15: RE24.

Figure 16: MO-LQR2.

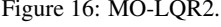

Figure 17: MO-LQR3.

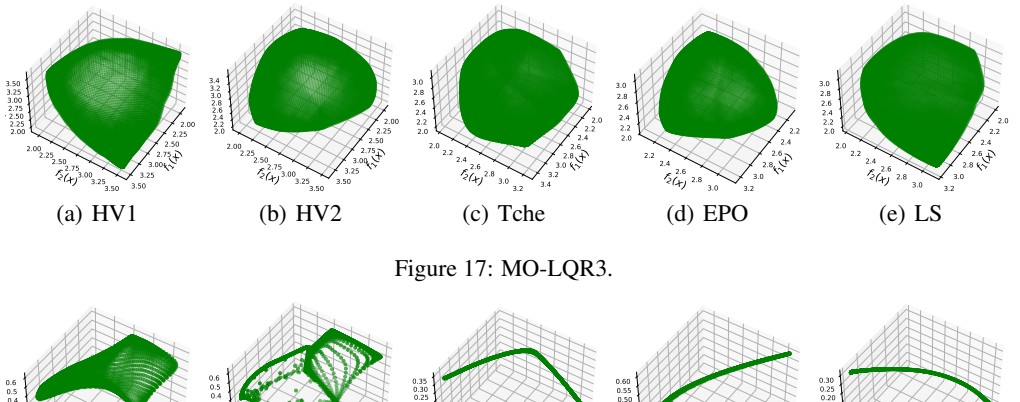

Figure 18: RE37.

Here, $\Omega$ denotes the region dominated by $\mathcal{F}^*$ with a reference point $r$, $\Omega = \{q \mid \exists p \in \mathcal{F}^* : p \preceq q \text{ and } q \preceq r\}$. $I_\Omega$ is the indicator function of $\Omega$. $\bar{c}_m$ is the volume of a m-D unit sphere, $\bar{c}_m = \frac{\pi^{m/2}}{\Gamma(m/2+1)}$. $c_m$ is a constant defined in the main paper, $c_m = \frac{\pi^{m/2}}{2^m \Gamma(m/2+1)}$.

Line 2 holds since it represents the integral of $\Omega$ expressed in polar coordinates, wherein the element $dv$ corresponds to the volume associated with a segment obtained by varying $d\theta$.

Line 3 calculates the infinitesimal volume of $dv$ by noticing the fact that the ratio of $dv$ to $\bar{c}_m$ is $\frac{\rho_\mathcal{X}(\theta)^m}{2\pi \cdot \pi^{m-2}}$. Line 4 is a simplification of Line 3. And Line 5 and 6 express the integral in its expectation form. $\qquad \square$

Table 6: The notation table.

| Variable | Definition |
|---|---|
| $x$ | The decision variable. |
| $n$ | The number of the decision variables. |
| $N$ | The number of samples. |
| $m$ | The number of objectives. |
| $\theta$ | The angular polar coordinate. |
| $\lambda(\theta)$ | An $m$-dimensional preference vector. |
| $\beta$ | The model parameter. |
| $y^{\text{nadir}}/y^{\text{ideal}}$ | The nadir/ideal point of a given MOO problem. |
| $\mathcal{F}^*$ | The Pareto front, which is set of all Pareto non-dominated solutions. |
| $\mathcal{H}_r(A)$ | The hypervolume of set $A$ w.r.t a reference . |
| $\mathcal{S}_+^{m-1}$ | The ($m$-1)-D positive unit sphere. |

## B.3 Proof of $\rho_{\mathcal{X}}(\theta)$ as a Max-Min Problem

We provide the proof of the following equation (Equation (6) in the main paper) in this subsection.

$$\rho_{\mathcal{X}}(\theta) = \max_{x \in \mathcal{X}} \rho(x, \theta) = \max_{x \in \mathcal{X}} \min_{i \in [m]} \{\frac{r_i - f_i(x)}{\lambda_i(\theta)}\}.$$

*Proof.* Let $x^*$ be one of the optimal solutions of Problem $\max_{x \in \mathcal{X}} \rho(x, \theta)$. To begin, we define the attainment surface $\mathcal{S}_{\text{attain}}$, as detailed in [31], utilizing a reference point $r$. The sets of Pareto solutions and weakly Pareto solutions are denoted as $\mathcal{F}^*$ and $\mathcal{F}^*_{\text{weak}}$, respectively. Then, $\mathcal{S}_{\text{attain}}$ is defined as,

$$\mathcal{S}_{\text{attain}} = \mathcal{F}^* \cup \{p \mid p \preceq r, \ p \in \mathcal{F}^*_{\text{weak}}\}. \tag{21}$$

We denote $P(\theta)$ as the intersection point of the ray from the pole $r$ along angle $\theta$ and the attainment surface $\mathcal{S}_{\text{attain}}$. $\rho_{\mathcal{X}}(\theta)$ is the distance from the reference point $r$ to the intersection point $P(\theta)$. There are two cases, $x^*$ is a Pareto solution or a weakly Pareto solution. Else, by contradiction, $f(x^*)$ can be improved in all objectives, $x^*$ cannot be a solution of Problem (6).

**When $x^*$ is Pareto optimal.** In such case, we should prove that $f(x^*) = P(\theta)$. If $x^* \neq P(\theta)$, then there exist at least one element $j$ such that, $\frac{r_j - f_j(\theta)}{\lambda_j(\theta)} \leq \frac{r_i - P_i(\theta)}{\lambda_i(\theta)}, \ \forall i = 1, \ldots, m$. This is a contradiction with $x^*$ is the optimal solution of Problem (6). So, $x^* = P(\theta)$.

**When $x^*$ is weakly Pareto optimal.** In such case, $f(x^*)$ does not necessary equals to $P(\theta)$. In such case, since $x^*$ is the solution of Problem (6), we have that there exist at least one index $j$, where $j = \arg \min \frac{r_j - f_j(x^*)}{\lambda_j(\theta)}$ such that $\frac{r_j - f_j(x^*)}{\lambda_j(\theta)} = \frac{r_i - P(\theta)}{\lambda_i(\theta)}, \ i = 1, \ldots m$. In such a case, $dist(P(\theta), r) = \frac{r_j - f_j(x^*)}{\lambda_j(\theta)}$. $\qquad \square$

## B.4 Proof of Proposition 2

This subsection provides the proof for Proposition 2, which builds the relationship between a polar angle $\theta$ and the corresponding solution of Problem (6).

*Proof.* There are two cases for $x^*$. $x^*$ is Pareto optimal or $x^*$ is weakly Pareto optimal. When $x^*$ is neither Pareto optimal nor weakly Pareto optimal, there exists a solution $x'$ which is better than $x^*$ for all objectives. In such case, $x^*$ is not a solution for Problem (6), which is a contradiction.

**When $x^*$ is Pareto optimal.** Since we have $\rho_{\mathcal{X}}(\theta) = \frac{r_i - f_i(x^*)}{\lambda_i(\theta)}$, which indicates that for any other solution $x'$, there exist at least one index $j$ such that, $\frac{r_j - f_j(x')}{\lambda_j(\theta)} \leq \rho_{\mathcal{X}}(\theta)$, then $x'$ is not the optimal solution of Problem (6). As a result $x^*$ is the only solution of Problem (6), $\mathcal{X}_\theta = \{x^*\}$.

**When $x^*$ is weakly Pareto optimal.** There can exist one solution $x'$ such that, $x_i' \neq x_i^*$ for some $i$ and therefore, $x' \in \mathcal{X}_\theta$. As a result, we can conclude that, $x^* \in \mathcal{X}_\theta$. $\qquad\square$

## B.5 Case of a Disjointed Pareto Front

In order to gain a more thorough comprehension of our approach to optimizing loss functions for Pareto set learning (PSL), we investigate a scenario where the Pareto front is disjointed. In such a scenario, it is noted that the preference vector still has an intersection point with the attainment surface (defined in Equation (21)), as illustrated by the blue curve in Figure 19. Equation (6) now measures the volume within the attainment surface and the reference point $r$, which is just the hypervolume of a disjointed Pareto front $\mathcal{H}_r(\mathcal{F}^*)$.

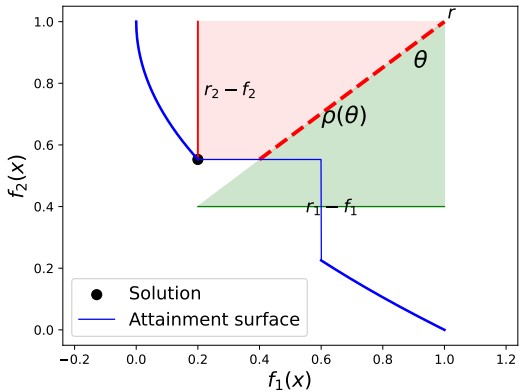

Figure 19: Case of a disjointed Pareto front.

For a disjointed Pareto front, the quantity $\rho_\mathcal{X}(\theta)$ denotes the distance between $r$ and the attainment surface associated with angle $\theta$. Specifically, in Figure 19, the black dot represents the solution for this scenario. The integral of the distance function $\rho_\mathcal{X}(\theta)$ still returns the hypervolume of a disjointed Pareto front, which satisfies our purpose in this paper.

However, disjointed Pareto fronts in Pareto set learning overemphasize boundary solutions which may result in unpredictable outcomes. For disjointed Pareto fronts, it is recommended to adaptively adjust the preference distribution (which is set to be uniform in our experiments).

## B.6 Pareto Front Hypervolume Calculation (Type2)

In this subsection, we define region $A$ as the set of points dominating the Pareto front,

$$A = \{q \mid \exists p \in \mathcal{F}^* : p \leq q \text{ and } q \geq p^{\text{ideal}}\}. \tag{22}$$

To ensure consistency with the notation used in the main paper, we use the notation $\Lambda(\cdot)$ to represent the Lebesgue measure of a set. From a geometric perspective, as illustrated in Figure 20, it can be observed that:

$$\Lambda(A) + \mathcal{H}_r(\beta) = \prod_{i=1}^{m}(r_i - y_i^{\text{ideal}}). \tag{23}$$

The volume of $A$ can be calculated in a polar coordinate as follows,

$$\Lambda(A) = c_m \int_{(0,\frac{\pi}{2})^{m-1}} \overline{\rho}_\mathcal{X}(\theta)^m d\theta, \tag{24}$$

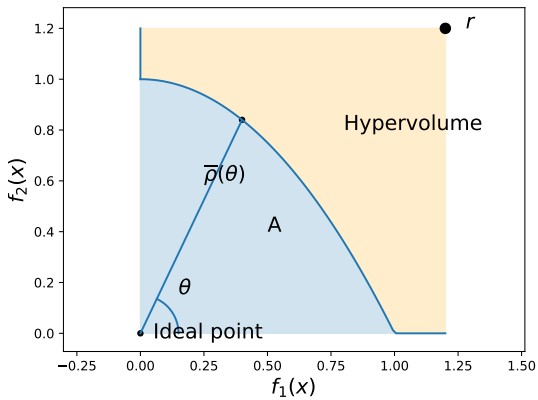

Figure 20: The hypervolume calculation (Type2).

where $c_m$ is a constant and $\overline{\rho}_\mathcal{X}(\theta)$ represents the distance from the ideal point to the Pareto front at angle $\theta$. This distance function $\overline{\rho}_\mathcal{X}(\theta)$ is obtained by solving the optimization problem assuming that any radius from $\theta$ intersects with the Pareto front.

**Problem 1.**

$$\overline{\rho}_\mathcal{X}(\theta) = \min_{x \in \mathcal{X}} \overline{\rho}_\mathcal{X}(\theta, x) = \min_{x \in \mathcal{X}} \max_{i \in [m]} \left\{ \frac{f_i(x) - y_i^{ideal}}{\lambda_i(\theta)} \right\}, \quad \theta \in (0, \frac{\pi}{2})^{m-1}. \tag{25}$$

The relationship between preference $\lambda$ and the polar angle $\theta$ is as follows:

$$\begin{cases} \lambda_1(\theta) = \sin\theta_1 \sin\theta_2 \ldots \sin\theta_{m-1} \\ \lambda_2(\theta) = \sin\theta_1 \sin\theta_2 \ldots \cos\theta_{m-1} \\ \ldots \\ \lambda_m(\theta) = \cos\theta_1. \end{cases} \tag{26}$$

Combining Equation (24) and (25) implies that $\overline{\mathcal{H}}_r(\beta)$ can be estimated as an expectation problem,

$$\overline{\mathcal{H}}_r(\beta) = \prod_{i=1}^{m}(r_i - y_i^{\text{ideal}}) - \frac{1}{m}c_m \mathbb{E}_{\theta \sim \text{Unif}(\Theta)}[\overline{\rho}_{\mathcal{X}}(x_\beta(\theta), \theta)^m]. \tag{27}$$

## B.7  Proof of Proposition 3

*Proof.* It can be observed that Equation (6) in the main paper implies the following equation,

$$-\rho(x, \theta) = \max_{i \in [m]} \left\{ \frac{f_i(x) - r_i}{\lambda_i(\theta)} \right\}. \tag{28}$$

When all objectives $f_i$'s are convex, function $-\rho(x, \theta)$ is also convex yet non-smooth, and hence $\rho(x, \theta)$ is concave. When $f_i$'s are differentiable, $-\rho(x, \theta)$ possesses a natural subgradient denoted as $d$ that is formulated as $d = \frac{\partial f_j(x)}{\partial x} \frac{1}{\lambda_i(\theta)}$, where $j = \arg\max_{i \in [m]}\{\frac{f_i(x) - r_i}{\lambda_i(\theta)}\}$. The subgradient $d$ can be iteratively updated to converge on the global optima of $\rho_{\mathcal{X}}(\theta)$ in a $\mathcal{O}(1/\epsilon^2)$ rate, as described in [48, 49].

When all objectives $f_i$'s are quasi-convex, $-\rho(x, \theta)$, which is a point-wise max of quasi-convex functions, is quasi-convex. And, hence $\rho(x, \theta)$ is quasi-concave. $\square$

## B.8  Proof of $\rho_\beta(\theta)$ is Quasi-Concave w.r.t. $x$

*Proof.* Proposition 3 rigorously demonstrates that the function $-\rho(x, \theta)$ is convex for any given value of $\theta$. Furthermore, consider the function $h(x) : \mathbb{R} \to \mathbb{R}$ which may be defined as follows,

$$h(u) = \begin{cases} u^m & \text{if } u \geq 0 \\ u & \text{otherwise} \end{cases}. \tag{29}$$

It is clear $h(x)$ is a non-decreasing function, and $g(x) = -\rho_\beta(x) = h \circ (-\rho(x, \theta))$. Since $(-\rho(x, \theta))$ is convex, then, for any $\alpha$, the set $S_\alpha(-\rho(x, \theta))$, as defined as follows, is convex.

$$S_\alpha(-\rho(x, \theta)) = \{x | -\rho(x, \theta) \leq \alpha\}. \tag{30}$$

Let $\gamma = h(\alpha)$. Then for any $\gamma$, the set $S_\gamma(h \circ (-\rho(x, \theta)))$, which equals to $S_\alpha(-\rho(x, \theta))$, is convex. This indicates that $h \circ (-\rho(x, \theta))$ is quasi-convex, and as a result $\rho_\beta(\theta)$ is quasi-convex w.r.t. $x$. $\square$

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

## B.10 Upper Bound of $\rho_\mathcal{X}(\theta)$

255 In this subsection, we prove that the distance function $\rho_\mathcal{X}(\theta)$ is bounded by the following inequality,

$$\rho_\mathcal{X}(\theta) \leq Bm^{1/2}, \tag{31}$$

256 when $r_i - f_i(x) \leq B, \ \forall x \in \mathcal{X}, \forall i \in [m]$ and $\|\lambda(\theta)\| = 1$.

257 *Proof.* We show that the following inequalities hold,

$$\rho_\mathcal{X}(\theta) \leq \max_{x \in \mathcal{X}, \|\lambda(\theta)\|=1} \left( \min_{i \in [m]} \{ \frac{r_i - f_i(x)}{\lambda_i(\theta)} \} \right)$$

$$\leq \max_{\|\lambda(\theta)\|=1} \left( \min_{i \in [m]} \{ \frac{B}{\lambda_i(\theta)} \} \right) \tag{32}$$

$$\leq \frac{B}{m^{-1/2}} = Bm^{1/2}.$$

258 The transition from line one to line two is due to the fact that the inequality $r_i - f_i(x) \leq B$
259 holds for all $x \in \mathcal{X}$ and for all $i \in [m]$. The transition from line two to line three is
260 $\max_{\|\lambda(\theta)\|=1} \left( \min_{i \in [m]} \{ \frac{B}{\lambda_i(\theta)} \} \right)$ is an optimization problem under the constraint $\|\lambda(\theta)\| = 1$.
261 The upper bound for this optimization is when $\lambda_i = \ldots = \lambda_m = m^{-1/2}$. $\qquad\square$
262 Let $\mathcal{Z}(\theta) = c_m \rho_\mathcal{X}(\theta)^m$, as a corollary, $\mathcal{Z}(\theta) \leq c_m B^m m^{m/2}$.

## B.11 Gradients of HV-PSL

264 In this subsection, we present the analytical expression for $\nabla_\beta \mathcal{H}_r(\beta)$ to ensure completeness. The
265 gradient for PSL-HV1 can be computed using the chain rule, which yields:

$$\nabla_\beta \mathcal{H}_r(\beta) = \begin{cases} mc_m \mathbb{E}_{\theta \sim \mathrm{Unif}(\Theta)}[\rho(x_\beta(\theta), \theta)^{m-1} \underbrace{\frac{\partial \rho(x_\beta(\theta), \theta)}{\partial x_\beta(\theta)}}_{1 \times n} \underbrace{\frac{\partial x_\beta(\theta)}{\partial \beta}}_{n \times d}], & \rho(x_\beta(\theta), \theta) \geq 0. \\[4ex] c_m \mathbb{E}_{\theta \sim \mathrm{Unif}(\Theta)}[\underbrace{\frac{\partial \rho(x_\beta(\theta), \theta)}{\partial x_\beta(\theta)}}_{1 \times n} \underbrace{\frac{\partial x_\beta(\theta)}{\partial \beta}}_{n \times d}], & \text{Otherwise.} \end{cases} \tag{33}$$

266 The gradient of PSL-HV2 can be calculated by,

$$\nabla_\beta \mathcal{H}_r(\beta) = -mc_m \mathbb{E}_{\theta \sim \mathrm{Unif}(\Theta)}[\overline{\rho}_\mathcal{X}(x_\beta(\theta), \theta)^{m-1}] \underbrace{\frac{\partial \overline{\rho}_\mathcal{X}(x_\beta(\theta), \theta)}{\partial x_\beta(\theta)}}_{1 \times n} \underbrace{\frac{\partial x_\beta(\theta)}{\partial \beta}}_{n \times d}. \tag{34}$$

## B.12 Relationship between Hypervolume and Decomposition based Multiobjective Optimization

269 In this subsection, we will explore the fundamental relationship between hypervolume-based
270 and decomposition-based multiobjective optimization. Prior to our study, it was commonly ac-
271 knowledged that there were three primary multiobjective optimization methods: Pareto-based [9],
272 hypervolume-based [30], and decomposition-based methods [8].

The present paper yields a result by establishing a correlation between hypervolume and decomposition-based approach in scenarios where the number of preference $\lambda(\theta)$ is considerably high. Previous methods mainly consider two decomposition functions, namely linear scalarization and Tchebycheff. Actually, we only need to make two modifications for the classical decomposition-based method in [8],

1. Sampling the polar angles $\theta^{(i)}$ from $S_+^{m-1}$.

2. For each sampled angle $\theta^{(i)}$, maximizing the scalarization function $\rho_{\mathcal{X}}(\theta^{(i)}) = \max_{i \in [m]} \{ \frac{r_i - f_i(x)}{\lambda_i(\theta^{(i)})} \}$.

Subsequently, upon optimizing each scalarization function, it becomes feasible to constrain the deviation between the empirical mean of $c_m \rho_{\mathcal{X}}(\theta^{(i)})^m$ and the hypervolume of the Pareto front to a small value with a high level of certainty. This is elaborated by Equation (9) in the main manuscript.