# OpenReview forum: "Hypervolume Maximization: A Geometric View of Pareto Set Learning"
_NeurIPS.cc/2023/Conference — NeurIPS 2023 poster_

### Official Review · Reviewer_jfi6 · 2023-07-06

**Soundness:** 3 good
**Presentation:** 3 good
**Contribution:** 2 fair
**Rating:** 7
**Confidence:** 3

**Summary:**

This manuscript introduces a method for multi-objective optimization and mainly contributes to modeling the Pareto set. Compared to previous methods which usually focus on limited and discrete Pareto solutions, the method in this manuscript is capable of modeling the full Pareto set in a continuous way. They also proposed a connection between the complete Pareto set and the related hypervolume, enabling a convergence analysis of the hypervolume as a novel metric for Pareto set learning. Also, this method bridges the Pareto solutions and their representation in a polar coordinate system. The authors also evaluate the proposed method extensively in a set of tasks.

**Strengths:**

1. The presentation is clear. The visualization is intuitive and helps to understand.
2. The experiment results look convincing. I do expect some deeper tasks though, besides these simple tasks.

**Weaknesses:**

1. The text on the figures is too small to read. Maybe considering to add a description in the caption.

**Questions:**

1. I wonder how this method generalizes in a more general multi-task learning context - for example, the accuracy of the network and the total norm of it (performance vs regularization).
2. I wonder how this method compares to Pareto exploration methods like [1], which also explores Pareto set in a continuous way.

[1] Ma, Pingchuan, Tao Du, and Wojciech Matusik. "Efficient Continuous Pareto Exploration in Multi-Task Learning." International Conference on Machine Learning. PMLR, 2020.

---

> ### Author Rebuttal · Authors · 2023-08-06
>
> We sincerely appreciate your valuable feedback. We hope that our following response can address your concerns.
>
> **W1. The text on the figures is too small to read. Maybe considering to add a description in the caption.**
>
> Thank you for your guidance. In the revised edition, we have implemented a larger font size ($\geq 16$) for all labels and tick marks in all figures.
>
>
> **Q1. Multi-Task Learning: I wonder how this method generalizes in a more general multi-task learning context - for example, the accuracy of the network and the total norm of it (performance vs regularization).**
>
> We recognize the importance of applying our proposed framework to **large-scale multi-task learning** situations. To achieve this, one potential approach is to utilize task-specific neural networks to focus on learning the trade-off objective. Concurrently, a backbone network can be employed to learn shared parameters that benefit all tasks.
>
> This effective technique was originally introduced in prior literature [1]. Moreover, it is worth noting that even for complex problems like multi-objective neural combinatorial optimization [2] and multi-objective drug design using GFlowNet [3], the number of task-specific parameters can be managed at a lower level. Future work is to learn the **optimal network structure** that effectively balances the size of the network (as referred to in the context of **normalization**) while simultaneously maximizing accuracy.
>
>
> **Q2. Comparison to Pareto Exploration Method: I wonder how this method compares to Pareto exploration methods like [2], which also explores Pareto set in a continuous way. Ma et al, [4].**
>
> We appreciate your advice, and we intend to incorporate a discussion on Ma et al.'s method [4] in our revised paper. Their approach employs the Krylov subspace method for local exploration using first-order approximation, with the objective of obtaining approximate Pareto solutions in close proximity to a discovered Pareto solution. The discovered Pareto solutions are pre-computed using the methodology introduced by Lin et al. in PMTL [5].
>
> Due to the local approximation nature of their method for generating continuous solutions, the accuracy of approximating distant solutions diminishes when they are far from a discovered solution. Consequently, as depicted in Figures (2), (10), and (12) of Ma et al.'s work [4], the ability of their approach to approximate Pareto solutions decreases when the distance between the Pareto solution and the discovered solution increases.
>
> The proposed method in our work directly learns the mapping from the preference space, which corresponds to the ($m$-1)-dimensional unit sphere (where $m$ represents the number of objectives), to the complete set of Pareto solutions. As a result, the quality of the learned solution remains unaffected by the size of the neighborhood, unlike in Ma et al.'s method [4]. This characteristic ensures that our approach achieves superior overall quality compared to their approach.
>
> **Reference**
>
> [1]. Multi-Task Learning as Multi-Objective Optimization. NeurIPS 2018.
>
> [2]. Pareto Set Learning for Neural Multi-Objective Combinatorial Optimization. ICLR 2022.
>
> [3]. Multi-Objective GFlowNets. ICML 2023.
>
> [4]. Efficient Continuous Pareto Exploration in Multi-Task Learning." ICML. 2020.
>
> [5]. Pareto multi-task learning. NeurIPS 2019.

---

> > ### Comment · Reviewer_jfi6 · 2023-08-16
> >
> > I would like to thank the authors for the effort of answering the my questions. I personally found the authors' responses very convincing (explaining the high-and-low-level multi-objective problems). Also, I agree with the authors that [Ma et al. 2020] can only capture the local approximation by the nature of the algorithm they used. I held the positive opinion previous and I still do now. I will keep my score.

---

### Official Review · Reviewer_Urc8 · 2023-07-07

**Soundness:** 3 good
**Presentation:** 3 good
**Contribution:** 3 good
**Rating:** 6
**Confidence:** 4

**Summary:**

This paper presents a novel perspective of hypervolume maximization in Pareto learning. The proposed method is largely interpretable based on this perspective, and could outperform many peer competitors considering efficiency and solutions.

**Strengths:**

1. The hypervolume analysis from the polar coordinate perspective is elaborate and reasonable.
2. The proposed method achieved a generally better performance than compared evolutionary and Pareto learning algorithms.


**Weaknesses:**

1. The innovations are not emphasized. For example, if the benefit of the proposed method based on geometric analysis include improved efficiency compared with other Pareto learning methods, it is suggested to be emphasized.
2. The details of the proposed method is inadequate. As the relationship between theoretical reasoning and methodology is obliquitous, the method is vulnerable to ignorance.
3. Comparisons and discussions may have missed some important Pareto learning techniques like hypernetworks. Contrasts or superiority of the proposed method are deficient.


**Questions:**

1. Does the proposed method introduce any type of preferences between objectives?
2. Does this method incorporate multiple neural models for joint learning?
3. What is the crucial determinant of the improved efficiency regarding this method?
4. How does scalability to large-scale problems embody in this work, typically compared with EPO-based approaches?


**Limitations:**

The limitations on solution quality is clearly mentioned.

---

> ### Author Rebuttal · Authors · 2023-08-06
>
> We sincerely thank you for the valuable comments on our work. We hope to address your concerns with the following answers.
>
> **W1. The innovations are not emphasized. For example, if the benefit of the proposed method based on geometric analysis includes improved efficiency compared with other Pareto learning methods, it is suggested to be emphasized.**
>
> - The first and main **innovation** of this work is its **error analysis**. While several Pareto learning methods exist (e.g., [1-3]), they lack theoretical analysis, and the properties of the learned Pareto model remain unclear. We establish a connection between PSL and hypervolume maximization, an important **"geometric"** quantity in multi-objective optimization (MOO), and provide convergence analysis for PSL with respect to HV maximization. This connection and the error bounds in this paper help us better understand PSL.
> - The second **innovation** of our proposed model is its ability to generate a Pareto solution that precisely aligns with the preference vector, enabling the localization of a Pareto solution from a **"geometric"** standpoint (as discussed in Property 2). In other words, our simple yet efficient method can produce an "exact" Pareto solution, as defined in EPO [3-4]. It's worth noting that our approach is 100 times faster than the EPO-based PSL method [3-4].
>
> We sincerely thank your advice, and we will make our motivations more clear in the next version.
>
> **W2. Details of the Proposed Method: The details of the proposed method is inadequate. As the relationship between theoretical reasoning and methodology is obliquitous, the method is vulnerable to ignorance.**
>
> We thank the reviewer for the suggestion. We will summarize our method in an **algorithm block** in the next revision and highlight the connection between a step in the algorithm and the corresponding quantity in our analysis.
>
> **W3. Comparisons and discussions may have missed some important Pareto learning techniques like hypernetworks. Contrasts or superiority of the proposed method are deficient.**
>
> The proposed framework is model-agnostic and orthogonal to techniques such as using a hypernetwork. For simplicity and theoretical analysis, the paper employs a fully-connected network. We believe that integrating hypernetworks into our model is straightforward by applying the optimization strategy described in Eq. (7) of the main paper.
>
> **Q1. Does the proposed method introduce any type of preferences between objectives?**
>
> In our Pareto neural model, "preferences" serve as inputs. Proposition 2 establishes the relationship between the preference vector $\lambda$ (expressed in polar coordinates) and the corresponding Pareto solution. Once the Pareto model is trained, it can produce a new (approximate) Pareto solution based on this new preference vector.
>
> **Q2. Does this method incorporate multiple neural models for joint learning?**
>
> Our current implementation utilizes a single neural network. The extension of the proposed method with joint learning is left as future work.
>
> **Q3-4. What is the crucial determinant of the improved efficiency regarding this method? How does scalability to large-scale problems embody in this work, typically compared with EPO-based approaches?**
>
> In short, our method enhances efficiency compared to the EPO-based approach by **eliminating** the need to calculate and operate on gradient vectors of **all** ($m$) objectives. Instead, we update the model **solely** based on the gradient of the argmin index in Eq. (6). It is worth noting that calculating a single gradient vector through backward propagation can be time-consuming in the default PyTorch implementation. As a result, the proposed approach becomes even more efficient than the EPO-based approach when applied to large-scale problems.
>
> To apply the proposed method to large-scale problems, we can utilize a backbone network that provides benefits across all objectives. By employing a small subset of networks, optimization can be performed independently for task-specific objectives, as described in [5]. This strategy allows us to update only the task-specific parameters using PSL. Our next plan is to incorporate this strategy into the proposed approach and apply it to large-scale problems.
>
> We hope our response adequately addresses your concerns, more or less. We appreciate any further comments or concerns you may have regarding our paper.
>
>
> Reference
>
> [1]. Multi–Objective Reinforcement Learning with Continuous Pareto Frontier Approximation. AAAI 2015.
>
> [2]. Multi-objective Reinforcement Learning through Continuous Pareto Manifold Approximation. JAIR, 2016.
>
> [3]. Learning the Pareto Front with Hypernetworks. ICLR, 2021.
>
> [4]. Gradient Descent with Controlled Ascent in Pareto Optimization. ICML 2020.
>
> [5]. Multi-Task Learning as Multi-Objective Optimization. NeurIPS 2018.
>
> [6]. Learning a Neural Pareto Manifold Extractor with Constraints. UAI, 2022.

---

> > ### Comment · Reviewer_Urc8 · 2023-08-15
> >
> > Dear authors: Thank you for your detailed responses and additional experiments, which have made the paper appear more comprehensible and rigorous. Generally, I have no more inquiries regarding this paper.

---

### Official Review · Reviewer_Kyce · 2023-07-07

**Soundness:** 3 good
**Presentation:** 3 good
**Contribution:** 3 good
**Rating:** 7
**Confidence:** 2

**Summary:**

This paper proposes a geometric perspective for Pareto set learning by establishing an equivalence between learning the complete Pareto set and hypervolume maximization. A theoretical analysis is provided to examine the gap between the estimated hypervolume and the true hypervolume of the Pareto set and empirical studies show its effectiveness.

**Strengths:**

1. This paper presents the equivalence between Pareto set learning and hypervolume maximization
2. Theoretical analysis of the gap between the estimated hypervolume and the true hypervolume of the Pareto set leads to better understanding in practice.

**Weaknesses:**

The empirical studies need to involve more problems, e.g., those from the popular benchmarks DTLZ and WFG, which have been shown to represent some difficult properties of real-world problems.

**Questions:**

How are the parameters of the experiment set? Is the complexity of the Pareto front taken into account when selecting test problems in the experiments?

**Limitations:**

It is suggested to analyze the effectiveness of the proposed learning method on problems with irregular Pareto front.

---

> ### Author Rebuttal · Authors · 2023-08-06
>
> We sincerely appreciate your valuable feedback. We hope that we can address your concerns with the following answers more or less.
>
> **W1. More Problems: The empirical studies need to involve more problems, e.g., those from the popular benchmarks DTLZ and WFG, which have been shown to represent some difficult properties of real-world problems.**
>
> Thank you for your suggestion. We have considered "more problems", e.g., the real-world problems mentioned in [6], which are four-objective designing problems. Detailed results can be found in the **general response** and the attached **one-page PDF**. These empirical results show that our method demonstrates strong potential as a multi-objective optimization approach.
>
> The benchmark problems in DTLZ and WFG were primarily intended for evolutionary multiobjective algorithms (EMOAs). Since our method is gradient-based, these problems are still challenging for PSL (and other PSL methods) due to multiple local Pareto fronts. We acknowledge this limitation around line 346 of the main paper. In response to that, we will provide a more comprehensive comparison of the advantages and disadvantages of the proposed method in the revised version.
>
>
> **Q1. Experiment Setting: How are the parameters of the experiment set? Is the complexity of the Pareto front taken into account when selecting test problems in the experiments?**
>
> The experimental setup is standard. All experiments are conducted using Python, utilizing the PyMoo and PyTorch libraries. Our experiments did not rely on GPUs. The neural model employed for these experiments is a 4-layer ReLU/Sigmoid network, and the optimizer used is SGD, with a clipping norm of 2.0. The batch size was set to 256.
>
> We also provide new results on the **effect of network structures** as the 3rd experiment in the **general response**.
>
> For your second question, in our paper, we have considered various "complexity levels" of the Pareto front, including convex examples such as ZDT1, concave examples like ZDT2, as well as real-world problems (with 3/4 objectives) with unknown Pareto front shapes, such as Four Bar Truss Design, Rocket Injector Design, and multi-objective linear quadratic regulator problems.
>
> **L1. Problems with Irregular Pareto Front: It is suggested to analyze the effectiveness of the proposed learning method on problems with irregular Pareto front.**
>
> To answer this question, we categorize irregular Pareto fronts (PFs) into two distinct cases: **disjointed** PFs and **degenerated** PFs. For a degenerated PF, for simplicity, we assume the PF is ($m$-2)-D, where $m$ is the number of objectives. This classification of irregular Pareto fronts is adopted from Hua et al. [5].
>
> In terms of disjointed Pareto fronts, the theoretical expectation of estimated hypervolume in our paper matches the true hypervolume of the Pareto front. This fulfills the theoretical analysis for this work. The challenges related to disjointed PFs are discussed in the main paper (lines 165-168). Additionally, a detailed analysis of these challenges can be found in section B.5 and Figure 19 of the supplementary material.
>
> In the case of a degenerated ($m$-2)-D Pareto front, sampling preferences on an ($m$-2)-D subspace can reconstruct the entire front. Consequently, preference sampling on the original ($m$-1)-D sphere results in duplicated Pareto solutions, reducing the learning efficiency. As far as we know, handling an irregular Pareto front remains a challenging and unresolved topic in multi-objective optimization. The crucial aspect lies in estimating the Pareto front to enable wise sampling of preference vectors. We have identified this as a potential area for future research.
>
> We hope to address your concerns. If you have any further questions, please let us know.
>
> Reference
>
> [1]. Multi–Objective Reinforcement Learning with Continuous Pareto Frontier Approximation. AAAI 2015.
>
> [2]. Multi-objective Reinforcement Learning through Continuous Pareto Manifold Approximation. JAIR, 2016.
>
> [3]. Pareto Set Learning for Expensive Multi-Objective Optimization. NeurIPS 2022.
>
> [4]. Diversity-Guided Multi-Objective Bayesian Optimization With Batch Evaluations. NeurIPS 2020.
>
> [5]. A Survey of Evolutionary Algorithms for Multi-Objective Optimization Problems with Irregular Pareto Fronts. IEEE/CAA Journal of Automatica Sinica, 2021.
>
> [6] An easy-to-use real-world multi-objective optimization problem suite. Applied Soft Computing, 2020.

---

> > ### Comment · Reviewer_Kyce · 2023-08-18
> >
> > Thank you for the response, all my concerns have been well addressed.

---

### Official Review · Reviewer_ZpxS · 2023-07-19

**Soundness:** 3 good
**Presentation:** 4 excellent
**Contribution:** 3 good
**Rating:** 6
**Confidence:** 3

**Summary:**

The author presents a novel approach to multiobjective algorithms aimed at modeling the Pareto set using neural networks. The proposed approach in the manuscript allows for the direct modeling of the entire Pareto set, and it also establishes an equivalence between learning the complete Pareto set and maximizing the associated hypervolume. In this study, the results of the proposed approach on various benchmark problems and real-world problems are encouraging, which makes it a promising alternative to existing multiobjective algorithms.

**Strengths:**

- The authors establish a crucial equivalence between learning the complete Pareto set and maximizing the associated hypervolume, which facilitates the convergence analysis of hypervolume for Pareto set learning.
- The author provides a clear interpretation of Pareto set learning as a hypervolume maximization problem, establishing a theoretical connection between the results of Pareto set learning and the hypervolume.
- Also, this paper establishes a direct correspondence between specific preferences and the resulting Pareto solution within a polar coordinate system, enhancing the interpretability of the approach.
- Further, it incorporates essential techniques in hypervolume-based Pareto set learning (PSL) for modeling the entire Pareto set, as discussed in Section 4.3. In this study, the results obtained from applying the proposed approach to various benchmark problems and real-world scenarios are highly encouraging, indicating its potential as a viable alternative to existing multiobjective algorithms.
- Overall, the main strengths lie in developing a novel approach to multiobjective algorithms, which enables direct modeling of the entire Pareto set using neural networks and demonstrates promising results.

**Weaknesses:**

-The paper acknowledges that the proposed approach relies on gradient-based methods, which can result in finding solutions that are locally optimal rather than globally optimal when dealing with non-convex objectives. Additionally, the effectiveness of classical nonparametric techniques in practical applications is uncertain.
-The computational latency of the PSL approach is recognized as a significant obstacle in effectively handling large-scale problems. This limitation prevents the method from scaling well to very large problems, thereby limiting its usefulness in practical scenarios.
-The paper highlights that the finite set learned from classical methods may not accurately approximate the continuous manifold of the Pareto set, particularly when there are multiple objectives involved. As a result, the method may struggle to accurately represent the Pareto set, which could result in suboptimal solutions. The generalizability of the proposed framework to other problems is unclear.

**Questions:**

- The author utilizes a 4-layer fully connected neural network in their demonstration. Is the performance of the network influenced by its structure? Are there any anticipated performance differences when using alternative architectures?
- In Table 2, what is the analysis regarding the occurrence of 0 values for range and sparsity when employing PSL-LS with ZDT2?
- What are some examples of classical nonparametric techniques that can enhance the robustness of the method and address the limitations of gradient-based methods?

**Limitations:**

- One major limitation is the reliance on gradient-based methods, which can lead to locally optimal solutions for non-convex objectives. Consequently, the method may not always find the global optimal solution. To address this issue and enhance the method's statistical guarantees, the authors propose exploring classical nonparametric techniques to boost its robustness. It would be clearer if the author could list some examples of such techniques in the manuscript.
- Another limitation is the computational latency associated with the PSL approach, which poses a significant challenge in handling large-scale problems. As a result, the method's scalability to very large problems is limited, which restricts its practical applicability.
- Further, the paper highlights that the finite set obtained from classical methods may not accurately approximate the continuous manifold of the Pareto set, particularly when there are numerous objectives. Thus, the method may fail to precisely represent the Pareto set, potentially leading to suboptimal solutions.

---

> ### Author Rebuttal · Authors · 2023-08-06
>
> We are grateful for your constructive comments on our work. We hope that our following response can address your concerns more or less.
>
> **Q1. Is the performance of the network influenced by its structure? Are there any anticipated performance differences when using alternative architectures?**
>
> The performance is **indeed** affected by the network structure. We present **numerical results** for various network structures as the 3rd experiment in the **general response**, revealing that the **depth** of the net has **a minimal impact** on the results, while the **width** does play a significant role. Decreasing the network width to 64 adversely affects performance. This observation aligns with a theoretical study in [6].
>
> **Q2. In Table 2, what is the analysis regarding the occurrence of 0 values for range and sparsity when employing PSL-LS with ZDT2?**
>
> PSL-LS is limited to identifying **a single solution** (Figure 11(e)). Due to the concave nature of the Pareto front ($f_2 = 1 - {(f_1)}^2, 0 \leq f_1 \leq 1$) in ZDT2, the LS-based PSL method can only find an arbitrary endpoint ((1,0) or (0,1)) of the Pareto front.
>
> **Q3;L1. What are some examples of classical nonparametric techniques that can enhance the robustness of the method and address the limitations of gradient-based methods?**
>
> The limitation of finding only a locally optimal solution is inherent in gradient-based multiobjective approaches. In the current stage, achieving global optimality, both theoretically and practically, is still challenging for PSL.
>
> Our rough plan for applying “nonparametric techniques”' is to change the function space of $x_\beta()$ from neural models to a certain reproducing kernel Hilbert space (RKHS) $H$ with the associated kernel $k(\cdot, \cdot): \Theta \times \Theta \mapsto R$, and turn to a regularized optimization problem $ \min_{x \in H} \frac{c_m}{N} \sum_{i=1}^N [\rho(x(\theta^{(i)}), \theta^{(i)})^m ] + \frac{1}{2} \lambda_r \\|x\\|_H^2$, where $\\|x\\|_H$ is the RKHS norm of the function $x()$.
>
> The usage of the RKHS may cause a loss of representation power while allowing a finer analysis and more robust optimization.
> In short, due to the representer theorem of RKHS, $x()$ (in contrast to the neural model $x_\beta()$) is now convex w.r.t. the tunable parameters, and therefore helps the convergence of ``the gradient-based methods''; the inclusion of the regularization term $\frac12 \lambda_r \\|x\\|_{H}^2$ is also supposed to improve the convexity and the robustness of the optimization problem.
>
> We thank the reviewer for the question and will incorporate the explanation above into the next revision.
>
> **L2. Another limitation is the computational latency associated with the PSL approach, which poses a significant challenge in handling large-scale problems.**
>
> We acknowledge that searching for the entire Pareto set, an ($m$-1)-dimensional continuous manifold, is indeed more challenging and time-consuming compared to searching for a single or a finite number of solutions. Notably, there have been successful applications of Pareto set learning (PSL) in large-scale problems such as multiobjective neural combinatorial optimization [1] and drug design [2].
>
> To address **large-scale PSL problems**, a useful technique (proposed in [3]) is to learn a backbone network parameterized by $\beta^{(sh)}$, which benefits all objectives independently of the preference information. Additionally, separate networks parameterized by $\beta^{(i)}(\lambda)$ conditioned on the preference information are employed to learn the trade-off objectives, where $\lambda$ is an $m$-D preference vector. In such a way, the total number of parameters $\beta=[\beta^{(sh)}, \beta^{(1)}(\lambda), \ldots, \beta^{(m)}(\lambda)]$ can be maintained in a low level.
>
> Compared to the widely used EPO-based PSL method introduced in [4-5], our proposed method has already shown superior efficiency. Table 2 and Figure 8 provide evidence that our method is 100 times faster and produces higher-quality Pareto solutions. Consequently, when dealing with large-scale problems, the proposed method outperforms EPO-based PSL in terms of efficiency.
>
> **L3. Numerous objectives.**
>
> We present new experiment results and promising findings on many-objective problems ($m \geq 4$). For detailed discussions, please refer to the **general response** and the **attached PDF**.
>
> For many-objective problems, representing the entire Pareto set with a finite number of populations becomes increasingly difficult since the solutions space is very large. The proposed method, aiming to learn the full continuous  Pareto set, provides much more Pareto solutions than traditional MOEAs. We believe that the idea of the proposed method offers an alternative way of solving many-objective problems.
>
> We welcome any additional comments or suggestions you may have.
>
> Reference
>
> [1] Pareto Set Learning for Neural Multi-Objective Combinatorial Optimization. ICLR, 2022.
>
> [2] Multi-Objective GFlowNets. ICML, 2023.
>
> [3]. Multi-Task Learning as Multi-Objective Optimization. NeurIPS, 2018.
>
> [4]. Learning the Pareto Front with Hypernetworks. ICLR, 2021.
>
> [5]. Gradient Descent with Controlled Ascent in Pareto Optimization. ICML, 2020.
>
> [6] Any Deep ReLU Network is Shallow. Arxiv, 2023.

---

> > ### Comment · Reviewer_ZpxS · 2023-08-11
> > **rebuttal**
> >
> > The author well addressed the questions that arose and responded to comments with further experiments to alleviate the limitations of the previous version.

---

### Official Review · Reviewer_6icQ · 2023-07-20

**Soundness:** 3 good
**Presentation:** 3 good
**Contribution:** 4 excellent
**Rating:** 7
**Confidence:** 3

**Summary:**

The paper presents an algorithm to learn a continous approximation of the Pareto frontier in multi-objective reinforcement learning (MORL). The idea of the algorithm is to train a neural network to produce solutions such that their hypervolume is maximized.
The idea iteself is not new, but the proposed method overcome well-known hard limitations of MORL algorithms, most notably the use of the hypervolume to train the neural network.
To overcome this limitation, the authors propose an approximation of the hypervolume based on its computation in polar coordinates.


**Strengths:**

To the best of my knowledge, this is the first paper that overcomes the hard computation of the hypervolume as loss to train neural networks to produce continuous Pareto frontier approximation.
The hypervolume computation is very expensive for problems with 3 objectives or more, and is non-differentiable.
The authors propose an approximation that overcomes these limitations and achieves good results in the experiments.

The authors also provide proofs in the appendix and acknowledge limitations of their approach, which I highly appreciate.


**Weaknesses:**

The authors missed some fundamental related work that is highly relevant to their method.

- The idea of learning a continuous Pareto frontier was first proposed --even though without neural network-- by Pirotta et al., "Multi-objective reinforcement learning with continuous pareto frontier approximation".
- Proposition 1 is equivalent to say that the Pareto frontier achieves the highest hypervolume. This was already shown by [17], so a reference should be added.
- The problem formulated in Eq. (3) is a special case of the more general version by Parisi et al., Multi-objective Reinforcement Learning through Continuous Pareto Manifold Approximation (Eq. (2)). In your case, you are using the hypervolume as indicator to be maximized.
- It would be nice to stress how hard the computation of the exact hypervolume is. In particular, Friedrich et al., "Friedrich, T., Horoba, C., & Neumann, F. (2009). Multiplicative approximations and the hypervolume indicator" showed that it is a #P-hard problem and proposed an approximation.

**Questions:**

I understand that the main contribution of the paper are theoretical, but could you run some experiments with more than 3 objectives? The MO-LQR can be easily customized to have N objectives, and it would be interesting to have a short experiment to investigate the computational and time complexity (as well as the other metrics like hypervolume and sparsity) of all algorithms (yours and baselines) on varying the number of objectives (2, 3, 4, 5).

**Limitations:**

The authors discuss the limitation of their work.

---

> ### Author Rebuttal · Authors · 2023-08-06
>
> Thank you for your helpful feedback. We greatly appreciate it. In response, we have carefully considered your concerns and incorporated them into our revisions.
>
> **W1-2. Missing Highly Relevant Work: The idea of learning a continuous Pareto frontier was first proposed --even though without a neural network-- by Pirotta et al. [1]. Proposition 1 is equivalent to say that the Pareto frontier achieves the highest hypervolume in [5]. So a reference should be added.**
>
> We would like to acknowledge that the concept of learning a continuous Pareto frontier was originally proposed by Pirotta et al. [1], and we recognize the valuable discussion of Proposition 1 by [5]. In the revised version of our paper, we have ensured to appropriately cite their contributions.
>
> **W3. Problem Formulation: The problem formulated in Eq. (3) is a special case of the more general version by Parisi et al. [2] (Eq. (2)). In your case, you are using the hypervolume as an indicator to be maximized.**
>
> Thank you for bringing Parisi et al.'s [2] work to our attention. In their work, they primarily focus on three indicators: the accuracy indicator $I_U$, the covering indicator $I_{AU}$, and the mixed indicator $I = \beta_1 I_{AU} / I_U - \beta_2$. We will make sure to cite their contributions appropriately. Due to the seminal nature of Parisi et al.'s work [2] in this field, many approaches to Pareto modeling are influenced, to some degree, by their work.
>
> Nevertheless, our proposed approach provides significant and **unique contributions** beyond Parisi et al.'s [2] work. The key differences and contributions can be summarized as follows:
>
> - Our method, a **conditioned model**, facilitates mapping preferences to Pareto solutions, enabling the **convenient generation** of user-specific solutions with a single input preference. In contrast, their method does not possess this capability.
> - Our proposed method establishes a crucial connection between Pareto set learning and hypervolume, a key indicator in multi-objective optimization (MOO). In addition, our approach includes an error analysis of the gap with the true hypervolume.
>
> **W4. Computational Complexity and Approximation of the Exact Hypervolume.**
>
> We agree that determining the hypervolume of a finite set of $n$ solutions with $m$ objectives is indeed a recognized NP-hard problem with respect to $m$, as indicated in reference [3]. We also would like to mention that the current best asymptotic runtime for $n$ solutions and $m$ objectives is $O(n \log n+n^{m/2})$, as proposed in reference [4].
>
> On the other hand, our proposed approach significantly diverges from previous methods [3-5] by estimating the hypervolume through an expectation problem, as depicted in Eq. (5) and Eq. (6). As a result, our primary focus in this paper is on addressing the **statistical** approximation error.
>
> Fortunately, the statistical hypervolume approximation error can be effectively bounded by the empirical mean of Eq. (5) when the number of samples is large. In this paper, we offer two types of approximation bounds (Eq. (9) and (10) in the paper).
>
> **Q1. Problems with More Objectives.**
>
> To address many-objective problems ($m \geq 4$), we provide a comprehensive discussion in the 1st experiment in the **general response**. Visualization results can be found in the **attached PDF**. In short, the experiments demonstrate the **good scalability** of the proposed method when applied to many-objective problems. Given the challenge of representing a high-dimensional Pareto set with finite populations using traditional MOEAs, our proposed method, which learns the entire Paret set, provides a valuable tool for handling many-objective problems.
>
> If you have any additional remarks or questions, please feel free to let us know.
>
>
> **Reference**
>
> [1]. Multi–Objective Reinforcement Learning with Continuous Pareto Frontier Approximation. AAAI 2015.
>
> [2]. Multi-objective Reinforcement Learning through Continuous Pareto Manifold Approximation. JAIR, 2016.
>
> [3]. Multiplicative approximations and the hypervolume indicator. GECCO, 2019.
>
> [4]. S-Metric Calculation by Considering Dominated Hypervolume as Klee's Measure Problem. Evolutionary Computation, 2009.
>
> [5]. The Hypervolume Indicator Revisited: On the Design of Pareto-compliant Indicators Via Weighted Integration. EMO, 2007.

---

> > ### Comment · Reviewer_6icQ · 2023-08-12
> >
> > Thank you for your response, all my questions have been well-addressed.

---

### Official Review · Reviewer_FLs5 · 2023-07-25

**Soundness:** 3 good
**Presentation:** 3 good
**Contribution:** 3 good
**Rating:** 7
**Confidence:** 3

**Summary:**

This paper presents a novel approach to multiobjective algorithms that allows for the direct modeling of the entire Pareto set. The authors present a novel approach to Pareto set learning (PSL) from a geometric perspective, distinguishing it from existing methods that treat all preferences equally, resulting in a mere partial Pareto front. The contributions of this paper are as follows:
* Introducing a connection between preferences and their corresponding Pareto solutions, enabling the learning of the complete Pareto front.
* Proposing a novel geometric perspective for PSL, demonstrating the equivalence of Pareto set learning to Hypervolume maximization.
* Utilizing a neural network model to effectively approximate the entire Pareto set.
* The experimental results validate the superiority of the proposed method over baseline approaches.


**Strengths:**

* The paper presents a novel approach to multiobjective algorithms that allows for the direct modeling of the entire Pareto set.
* The authors establish an equivalence between learning the complete Pareto set and maximizing the associated hypervolume, which enables the convergence analysis of hypervolume for Pareto set learning.
* The theoretical foundation of the proposed method is well established, especially the derivation of the generalization gap between estimated and true hypervolumes.
* The proposed approach is evaluated on various benchmark and real-world problems, and compared to multiple state-of-the-art algorithms, and the results are promising.


**Weaknesses:**

* There is limited discussion regarding the design of the neural network structure, and the Machine Learning perspective could benefit from further elaboration.
* The discussion about the construction of the training data is lacking, which is crucial to understanding the similarities and differences between the training dataset and benchmark problems.
* The evaluation of the proposed model's PSL ability is solely compared with other PSL methods. It would be beneficial to also compare its optimization performance with other traditional methods, such as decomposition-based and hypervolume-based approaches.
* The focus of the tested problems is primarily on 2 or 3 objective optimization problems. However, it remains unclear if the proposed method can be generalized to handle more objectives.


**Questions:**

See Weaknesses.

**Limitations:**

The authors have adequately addressed the limitations.

---

> ### Author Rebuttal · Authors · 2023-08-06
>
> We sincerely appreciate your valuable feedback. We have taken your concerns, and we hope our following answers can address your concerns more or less.
>
> **W1. There is a limited discussion regarding the design of the neural network structure, and the machine learning perspective could benefit from further elaboration.**
>
> Thank you for the suggestion. In response, we conducted additional experiments in the $3rd$ experiment of the **general response** to examine the impact of network structure. Our findings suggest that the depth of the neural model has **minimal** influence on the generated quality, whereas the width **does** affect the results. Specifically, using a width of 64 leads to poor solutions. This finding aligns with the theoretical results discussed in previous research [3].
>
>
> **W2. The discussion about the construction of the training data is lacking, which is crucial to understanding the similarities and differences between the training dataset and benchmark problems.**
>
>  As shown in Figure 2, the studied Pareto set learning (PSL) problem involves learning the mapping function $x_\beta(\cdot)$ from a preference vector (in polar coordinates) and the corresponding Pareto solution. The considered Pareto neural model is a mapping function denoted as $x_\beta(\cdot): {[0, \pi/2]}^{m-1} \mapsto R^n$, where $m$ represents the number of objectives and $n$ denotes the dimensionality of the solution space. And therefore, according to Eq. (5), the "training data" (preference vectors) are sampled from the uniform distribution, $\text{Unif}({[0, \pi/2]}^{m-1})$.  This sampling method provides an unbiased estimation of the true hypervolume.
>
> **W3. The evaluation of the proposed model's PSL ability is solely compared with other PSL methods. It would be beneficial to also compare its optimization performance with other traditional methods, such as decomposition-based and hypervolume-based approaches.**
>
> Thank you for your feedback. We conducted "more" experiments, comparing the proposed method with traditional MOEAs, including **MOEA/D-Tche** [1], **SMS-EMOA** [2], and **NSGA2** [4]. Please refer to the **general response** and the **attached PDF** for details. In summary, our proposed method aims to generate the entire Pareto front, while traditional MOEAs rely on a finite population to approximate the Pareto set. We also observed that the advantages of the proposed Pareto set learning concept is particularly valuable for problems with many objectives (m ≥ 4) (in 1st experiment in general response), as approximating such a large Pareto manifold becomes increasingly challenging with only finite solutions.
>
> **W4. The ability to handle problems with more objectives.**
>
> In response to many-objective problems (m ≥ 4) issues, we have provided a comprehensive discussion about that in the 1st experiment in **general response** and the **attached PDF**. In short, our method displays **good scalability** when applied to many-objective problems. As the Pareto set of a 4-objective problem is large (a 3-D continuous manifold), approximating this set with finite solutions by traditional MOEAs poses increasing challenges. However, using the proposed method, it is easy to train a model to approximate the **entire** continuous Pareto set, which provides more alternative solutions for the user.
>
> We hope our response adequately addresses your concerns. If you have any additional questions or concerns, please feel free to let us know.
>
> Reference
>
> [1] MOEA/D: A Multiobjective Evolutionary Algorithm Based on Decomposition. IEEE TEVC, 2007.
>
> [2] SMS-EMOA: Multiobjective selection based on dominated hypervolume. EJOR, 2007.
>
> [3] Any Deep ReLU Network is Shallow. Arxiv, 2023.
>
> [4] A fast and elitist multiobjective genetic algorithm: NSGA-II. TEVC, 2002.

---

> > ### Comment · Reviewer_FLs5 · 2023-08-16
> > **Response to authors**
> >
> > Thanks to the authors for the revision. All my concerns have been addressed convincingly. I have updated my score accordingly.

---

### Author Rebuttal · Authors · 2023-08-07

**General response**

We sincerely appreciate the diligent efforts of all the reviewers. In response to their concerns, we have conducted three additional experiments.
- - -
> **1. Many-Objective Problem**

In our first experiment, we evaluate the performance of our proposed method on many-objective problems. We hope that this analysis can more or less address the concerns raised by **W4 for FLs5**, **Q1 for 6icQ**, **L3 for ZpxS**, and **W1 for Kyce**.
| Problem (obj=4)          | Indicator | Proposed PSL-HV2 (8000 sols) | NSGA2 (100 sols) |     PSL-EPO    |
|--------------------------|-----------|------------------------------|------------------|----------------|
| Car side impact design  (CSID)  | HV        | **1.88**                         | 1.66             |     1.08           |
|                          | Time      |     41.46(s)                 |    **26.36(s)**      |       19.63(m)         |
| Conceptual marine design (CMD) | HV        | **1.17**                         | 1.01             |     0.47*           |
|                          | Time      | 41.86(s)                     |     **26.68(s)**     |         19.44(m)       |

(*We have finetuned the parameters and learning rate of EPO using code from [5]. It does not work very well on CMD. )

We find that the proposed method improves the efficiency of learning the Pareto set for a large number of objectives ($m$=4) compared to traditional Multi-Objective Evolutionary Algorithms (MOEAs). In a 4-objective scenario, the Pareto set/front is a continuous 3-D manifold within an $n$/$m$-D space. The Pareto set is significantly larger than the problems considered in the main paper ($m$=2,3). Approximating this extensive Pareto set using finite solutions through traditional MOEAs presents a considerable challenge.

Taking NSGA2 [6] as a representative of conventional MOEAs, it takes approximately 26 seconds for NSGA2 to generate 100 solutions for these two problems. Our proposed method demonstrates remarkable scalability for 4-obj problems.  With a training time of just 40 seconds, we train a model which is able to (approximately) generate the **entire** Pareto set. The hypervolume of the entire Pareto set is greatly improved compared with NSGA2. The trained Pareto model has the theoretical ability to generate an infinite number of Pareto solutions, approximating the true 3-D Pareto manifold. In practice, we present results using 8000 solutions (they can be generated just very fast). For detailed results, please refer to the attached PDF.

We have observed that when applying our method to **many($m \geq 4$)-objective** problems, it **surpasses** the efficiency of EPO-based PSL. EPO-based PSL requires gradient calculations and operations for all objectives, leading to longer running times as reported in [4]. We are sincerely grateful to the reviewers for highlighting the importance of conducting experiments on many-objective problems. Based on our findings, we believe that our method holds great promise as a valuable tool for addressing many-objective problems.
- - -
> **2. Traditional MOEAs**

As suggested by **W3 for FLs5**, we have compared with more traditional MOEAs on a three-objective Rocket Injector Design problem [1]. Specifically, we consider the**MOEAD-Tche** [2] and **SMS-EMOA** [3]. Both MOEA/D-Tche and SMS-EMOA only generate a finite number (91) of solutions, as shown in the attached file. Figure 1 clearly demonstrates that the continuous Pareto front examined in this paper accurately represents the true Pareto front, showcasing its superior ability in dealing with a 3-obj problem. The running time of these three algorithms is presented in the following table.  As our method has the ability to generate an infinite number of solutions, its running time is **comparable** to that of MOEA-Tche (91 solutions) but significantly **outperforms** SMS-EMOA.

| Problem (obj=3)        | Indicator | MOEA/D-Tche | SMS-EMOA | Proposed (Training) |
|------------------------|-----------|-------------|----------|---------------------|
| Rocket Injector Design | Time      | 11.77s      | 8.35m    | 33.64s              |
---
> **3. Model Structure**

The last experiment studies the network structure effect raised by **W1 for FLs5**  and **Q4 for ZpxS**. We found that the **depth** of a ReLU network has **minimal impact**, while the **width does affect** the performance. Reducing the width to 64 generally leads to a decrease in performance. This observation aligns with a theoretical study in [7], and we plan to investigate further and gather additional results.
|                 | Rocket   Injector Design |       |          | Four Bar Truss   Design |       |          |
|-----------------|--------------------------|-------|----------|-------------------------|-------|----------|
| Network         | HV                       | Range | Sparsity | HV                      | Range | Sparsity |
| m-256-n         | 40.7                     | 0.7   | 2.2      | 11.9                    | 1.6   | 1.5      |
| m-256-256-n     | 40.5                     | 0.7   | 2.2      | 11.8                    | 1.5   | 1.8      |
| m-256-256-256-n | 40.6                     | 0.7   | 2.7      | 11.9                    | 1.6   | 1.5      |
| m-64-64-64-n    | 38.1                     | 0.3   | 1.3      | 11.8                    | 1.5   | 1.7      |

---

Reference

[1] An easy-to-use real-world multi-objective optimization problem suite. Applied Soft Computing, 2020.

[2] MOEA/D: A Multiobjective Evolutionary Algorithm Based on Decomposition, IEEE Transactions on Evolutionary Computation, 2007.

[3] SMS-EMOA: Multiobjective selection based on dominated hypervolume, European Journal of Operational Research, 2007.

[4] Learning a Neural Pareto Manifold Extractor with Constraints. UAI, 2022.

[5] https://github.com/dbmptr/EPOSearch.

[6] A fast and elitist multiobjective genetic algorithm: NSGA-II. TEVC, 2002.

[7] Any Deep ReLU Network is Shallow. arXiv, 2023.

---

### Decision · Program_Chairs · 2023-09-21

**Decision:**

Accept (poster)

**Comment:**

This work presents a novel approach to Pareto Set Learning (PSL). It shows the PSL is equivalent to hypervolume maximization, and shows how to efficiently optimize the hypervolume directly. The authors add a theoretical analysis and show their approach improves upon current methods on several benchmarks. Taking everything into account, it is an interesting, novel, and valuable paper that should be included in the following proceedings.